# A Unique Vadose Zone Model for Shallow Aquifers: the Hetao Irrigation District, China

Zhongyi Liu[1,2], Xingwang Wang[1], Zailin Huo[1*], Tammo S. Steenhuis[2*]

[1]Center for Agricultural Water Research in China, China Agricultural University, Beijing, 100083, PR China

[2]Department of Biological and Environmental Engineering, Cornell University, Ithaca, NY, USA.

Correspondence to: Zailin Huo (huozl@cau.edu.cn)

Tammo S. Steenhuis (tss1@cornell.edu)

**Abstract**

Rapid population growth is increasing pressure on the world water resources. Agriculture will require crops to be grown with less water. This is especially the case for the closed Yellow River basin necessitating a better understanding of the fate of irrigation water in the soil. In this manuscript, we report on a field experiment and develop a physically based model for the shallow groundwater in the Hetao irrigation district in Inner Mongolia, in the arid middle reaches of the Yellow River. Unlike other approaches, this model recognizes that field capacity is reached when the matric potential is equal to the height above the groundwater table and not by a limiting soil conductivity. The field experiment was carried out in 2016 and 2017. Daily moisture contents at 5 depths in the top 90 cm and groundwater table depths were measured in two fields with a corn crop. The data collected were used for model calibration and validation. The calibration and validation results show that the model-simulated soil moisture and groundwater depth fitted well. The model can be used in areas with shallow groundwater to optimize irrigation water use and minimize tailwater losses.

**Key words:** Hydrological model, Shallow aquifer, Equilibrium state, Soil moisture characteristic curve

**1 Introduction**

With global climate change and increasing human population, much of the world is facing substantial water shortage (Alcamo et al., 2007). The water crisis has caused widespread concern among public governmental officials and scientists (Guo and Shen, 2016; Oki and Kanae, 2006). Years of rapid population growth has squeezed the world water resources. The available fresh water per capita decreased from 13400 $m^3$ in 1962 to 5900 $m^3$ in 2014 (World Bank Group, 2019).

Water supply in China is especially stressed. When averaged over the whole country, available water per capita is at the water stress threshold of 1700 $m^3$ per year (Falkenmark, 1989; Brown and Matlock, 2011). It is even less in the arid to semi-arid Yellow river basin that produces 33% of the total agricultural production in China. To overcome water shortages in the Yellow river basin, crops are irrigated from surface and groundwater. This irrigation has directly changed the hydrology of the basin. While, 50 years ago, the semi-arid North China Plain had springs, shallow groundwater and rivers feeding the Yellow River, at the present rivers and springs have dried up where groundwater is used for irrigation (Yang et al., 2015a). At the same time, in the arid Inner Mongolia, along the Yellow River, the once deep groundwater is now within 3 m of the soil surface in the large irrigation projects

such as the Hetao irrigation district because of downward percolation of the excess irrigation water that has been
applied.

In the Yellow River basin, crop irrigation accounts for 96% of the total water use (Li et al., 2004). Due to the

increased demand for irrigation, the river has stopped flowing downstream for an average of 70 days per year
(Hinrichsen, 2002). Saving water upstream in Inner Mongolia by improved management practices mean that more
water will be available downstream (Gao et al., 2015).  In addition, the Hetao district is suffering from salinization
which leads to the land degradation (Guo et al., 2018; Huang et al., 2018) . Salinization is caused by upward
migration of water (and salt) from shallow groundwater table that leads to salt accumulation at the surface (Ren et
al., 2016; Yeh and Famiglietti, 2009). Designing improved management practices to save water and decrease
salinization can be achieved by field trials or with the aid of computer simulation mode measuring the fluxes. Field
trials are time consuming, expensive and only a limited set of water management practices can be investigated.
Models can test many management practices; however, the modeling results are often questionable because they
have not been validated under local field condition and have not been validated for the future conditions. A
combination of field experiments together with models has the benefits of both approaches with few negative effects.

Central to modeling irrigation management practices under shallow groundwater conditions (such as in the

Yellow river basin) is simulating the soil moisture content accurately (Batalha et al., 2018, Gleeson et al., 2016;
Jasechko and Taylor, 2015; Venkatesh et al., 2011a) because the moisture content plays a critical role in the growth
of crops (Rodriguez-Iturbe, 2000), groundwater recharge (Hodnett and Bell, 1986), upward movement of water to
the root zone in areas (Gleeson et al., 2016; Jasechko and Taylor, 2015; Venkatesh et al., 2011a; Batalha et al.,
2018). The latter is unique to shallow groundwater areas where the moisture content and thus the unsaturated
conductivity are high and where the drying of the surface soil sets up hydraulic gradient that causes the upward
capillary  movement from the shallow groundwater (Kahlown et al., 2005; Liu et al., 2016; Luo and Sophocleous,
2010; Yeh and Famiglietti, 2009). The upward moving water contains salt that is deposit in the root zone and at the
surface.
Modeling moisture contents

There is tendency with the ever increasing computer power, to include all processes and the highly

heterogeneous field conditions in hydrological models (Asher et al., 2015). In case of simulating moisture contents
these models become complex and often fully distributed in 3-D (Cui et al., 2017). Examples of these fully
developed models are HYDRUS (Šimůnek et al., 1998), SWAP (Dam et al., 1997) and MODFLOW (Mcdonald and
Harbaugh, 2003; Langevin, et al., 2017). These models have long run times when applied to scenarios simulations
for real world problems. In addition, calibration effort increases exponentially with the number of model parameters
(Rosa et al., 2012; Flint et al., 2002). This makes the use of the complex models for real time management and
decision support cumbersome where many model runs are needed (Cui et al., 2017).

To overcome the disadvantages of the full and completer models, computationally efficient surrogate models

have been developed to speed up the modeling process without sacrificing accuracy or detail. Surrogate models are
known under several names such as metamodels, reduced models, model emulators, proxy models and response
surfaces (e.g., Razavi et al., 2012a; Asher et al., 2015). The complex models we will call "full" or comprehensive
models.

Computational efficiency is the main reason for applying surrogate models in place of full models. Other

advantages of surrogate models are shortening the time needed for calibration; identifying insensitive and irrelevant
parameters in the full models (Young and Ratto, 2011). Most importantly, surrogate models allow investigating
structural model uncertainty (Matott and Rabideau, 2008). Finally, surrogate models might be able to deal better
with the self- organization of complex system prevalent in hydrology than the full models (Hoang et al., 2017). For
example, full models based on small scale physics (Kirchner, 2006) not necessarily can model the repetitive wetting
patterns observed in humid watersheds and for that reason.  Simple surrogate models often outperform their complex
counterparts in predicting runoff when a perched water table is present in sloping terrains (Moges et al, 2017; Hoang
et al 2017).

Surrogate models can be classified in two categories (Todini, 2007; Asher et al., 2015): data driven and

physically derived. Data driven surrogates analyze relationships between the data available and physically derived
surrogates simplify the underlying physics or reduce numerical resolution. In recent years, most emphasis in the
research literature has been data driven surrogate approaches (Razavi et al. 2012a). Relatively little research has
been published on physically derived approaches.  Despite its popularity, data-driven surrogates can be an inefficient
and unreliable approach to optimizing complex field situations especially when data is scarce such as in
groundwater systems  (Razavi et al. 2012b)  The physically derived surrogates overcome many of the limitations of
data-driven approaches and are therefore superior over data driven methods (Asher et al., 2015).
In the Yellow River basin various water accounting models have been developed to simulate the soil water
content and water fluxes (Xu, et al., 2012; Chen et al., 2014; Xue and Ren, 2017; Yang et al., 2017; Ren et al., 2019).
Numerical implementations are the finite element model HYDRUS-1D by Ren et al. (2016) and Luo and
Sophocleous (2010) and a finite difference model by Moiwo et al., (2010). Surrogate models for the North China
plain where the groundwater is more than 20 m deep have been published by Wang et al. (2001); Kendy et al (2003);
Chen et al. (2010); Ma et al. (2013);   Yang et al. (2015, 2017a,b); Li et al., (2017). In these models, the matric
potential is ignored, and the hydraulic potential is equal to the gravity potential and thus the gradient of the hydraulic
potential is unity (at least when it is expressed in head units). Under these conditions the water flux becomes
negligible when the soil reaches field capacity at -33 KPa (equivalent to -3.3 m in head units) at what point the
hydraulic conductivity becomes limiting. These models are not valid for irrigation projects along the Yellow river
with shallow groundwater because the matric potential cannot be ignored over the short distance between the water
table and the surface of the soil. Since the gravity and matric potential are of the same order, the  water moves either
down to the groundwater  or up from the groundwater to the root zone depending on the matric potential at the soil
(Gardner 1958; Gardener et al, 1970a,b). In summary, for shallow groundwater at less than 3.3 m from the surface
equilibrium is reached (i.e. fluxes negligible) when hydraulic gradient is zero (i.e., matric potential and gravity
potential add up to constant value) and thus not when the conductivity becomes limited at a matric potential of -33
KPa.
For the irrigation perimeters with shallow groundwater in the Yellow River basin, we could find only two
surrogate models developed by Xue et al. (2018) and Gao et al. (2017a, b). These two models do not consider the
dynamics of groundwater depth and matric potential. By including these dynamics more realistic predictions of
moisture contents and upward flow can be obtained and would give better results when extended outside the area
where they are developed for (Wang and Smith, 2004). The reason is that for areas with shallow groundwater,
evapotranspiration sets up hydraulic gradient that causes the upward capillary water movement to sustain the
evapotranspiration demands and crop water use (Kahlown et al., 2005; Liu et al., 2016; Luo and Sophocleous, 2010;
Yeh and Famiglietti, 2009).
Advantages of physically driven surrogates are particularly relevant groundwater studies where water tables are
simulated over entire large area as shown by Brooks et al. (2007). Despite this, Asher et al. (2015) poses that
physically driven methods have not been applied widely to groundwater problems and even fewer with the
interaction of moisture contents in the vadose zone which are key in salinization and plant growth of the many
cropped irrigated field in arid and semi-arid regions. In these water short areas it is extremely important to develop
models that show directions how to save water. The main objective of this study is, therefore, to develop a novel
surrogate model and validating this approach using experimental data collected in a field with shallow groundwater
with the ultimate goal is to save water in irrigation districts. In addition, sensitive and insensitive model parameters
were identified for simulating moisture content in shallow groundwater area to optimize future data collection
efforts. The experimental fields are located in the Hetao irrigation district, Inner Mongolia, China, where on two
maize fields, the moisture content and the groundwater table depth were measured over a two-year period.
The surrogate model developed is a one dimensional model simulating the moisture content in the root zone
using the groundwater depth and information of soil moisture characteristic curve. It can be easily adapted to field
scale by including the lateral movement of the regional groundwater. However, over short times, lateral movement
can be neglected in nearly level areas outside a strip of 5-100 m from the river (Saleh et al., 1989) such as deltas and
lakes (Dam et al., 1997; Kendy et al 2003).
**2 Materials and Methods**
**2.1 Study Area**
The Hetao Irrigation District (HID) is the third largest irrigation district of China. It covers an area of $1.12 \times 10^6$
ha of which half is irrigated (Xu et al., 2015). About 5 billion $m^3$ water are diverted from the Yellow River each year
(Xu et al., 2010). The primary irrigation method used is surface flood irrigation (Sun et al., 2013). The groundwater
table is very shallow ranging between 0.5 m to 3 m. The overall hydraulic gradient is 0.1-0.25‰ (Ren et al., 2018).
Soil salinization is serious, and the chemical composition of groundwater salinity mainly consists of NaCl, KCl,
$CaSO_4$. The Hetao District has a typical arid continental climate with high evaporation and low rainfall. The average
annual precipitation is 180 mm and the annual potential evapotranspiration is 2225 mm (Luan et al., 2018). The soil
is mainly alluvial deposits with a silty loam texture. It is frozen 5 to 6 months per year from late November to the
middle of May. Maize and wheat are the main food crops and sunflower is the main cash crop.
**2.2 Field experiment and data collection**
The experiment was carried out in Fenzidi, Bayannur city (41°9′N, 107°39′E) in the Hetao District in 2016 and
2017 (Fig.1). In 2016, the experiment was carried out separately in site A (about 3100 $m^2$) and site B (about 7000 $m^2$)
(Fig.1). In 2017, Field B was split into Fields B1 and B2 and experiments were carried out in these two fields. Field
B1 was about 3400 $m^2$ and B2 about 3600 $m^2$. Experimental fields were planted both years with maize. The sowing
dates were April 24, 2016 and May 13, 2017, respectively. The harvest date was October $1^{st}$ in both 2016 and 2017.
The plant growth stages are given in Table 1. The fields were flood irrigated three or four times during the heading
and filling stages starting in late June or early July (Table 2).

Precipitation, air temperature, relative humidity, sunshine duration and wind speed were collected from the

weather station at the experimental station. The reference evapotranspiration ($ET_0$) was calculated based on the
FAO-Penman-Monteith equation with the daily meteorological data (Allen et al., 1998). Precipitation and $ET_0$
during the growing season are shown in Fig. 2. The soil moisture was monitored daily in the top 90 cm using Hydra
Probe Soil Sensors (Stevens Water Monitoring System Inc., Portland, OR, USA) installed in both experimental
fields. Soil moisture was measured at 5 depths: 0-10 cm, 10-30 cm, 30-50 cm, 50-70 cm, and 70-90 cm. The sensors
were connected to data loggers and downloaded via wireless transmission. Calibration was conducted by oven
drying soil samples (Wang et al., 2018; Gao et al., 2017a). The groundwater depth was measured by piezometers
(HOBO Water Level Logger-U20, Onset, Cape Cod, MA, USA) recorded at 30 min intervals.

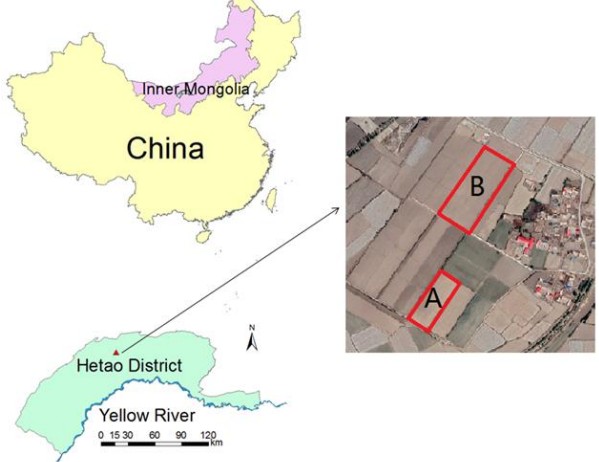


Figure. 1 Location of the field experiment in Hetao irrigation district. The blue line is the Yellow River.
Table 1
*Crop growth stage in 2016 and 2017 for corn growth on the Fenzidi experimental fields in the Hetao district*

| Year\Growth stage | seeding | jointing | heading | filling | maturing | harvesting |
|---|---|---|---|---|---|---|
| 2016 | 24-Apr | 25-May | 16-Jul | 6-Aug | 3-Sep | 1-Oct |
| 2017 | 13-May | 11-Jun | 18-Jul | 8-Aug | 5-Sep | 1-Oct |

Table 2
*Irrigation scheduling carried out at Fenzidi experimental fields in 2016 and 2017*

| Year | Field | Irrigation events | Date | Irrigation depth(mm) |
|---|---|---|---|---|
| | | First | July 13 | 115 |
| | A | Second | July 26 | 86 |
| | | Third | August 8 | 122 |
| 2016 | | First | June 23 | 57 |
| | | Second | July 13 | 119 |
| | B | Third | July 26 | 86 |
| | | Fourth | August 8 | 122 |
| | | First | July 13 | 153 |
| | B1 | Second | July 23 | 104 |
| | | Third | August 9 | 134 |
| 2017 | | First | July 13 | 165 |
| | B2 | Second | July 23 | 107 |
| | | Third | August 9 | 128 |

(a) Date        (b) Date

Figure. 2 Daily reference evaporation, ET0, and precipitation during the growing season in (a) 2016 and (b) 2017
Soil samples were collected in rings from the same five layers where moisture contents were measured and
used for determining soil physical properties including soil moisture at field capacity ($\theta_{fc}$), soil moisture at saturation
($\theta_s$), dry bulk density ($\rho$), and saturated hydraulic conductivity ($K_s$) (Table 3). For Fields A, B, B1 and B2, the
saturated hydraulic conductivity was determined by the constant head method. Field capacity was determined at - 33
kPa and bulk density was determined by oven drying and dividing by the volume of the ring. Soil texture of Fields A
and B were analyzed with the laser particle size analyzer (Mastersizer 2000, Malvern Instruments Ltd. United
Kingdom) in the laboratory and are shown in Table 4. The American soil texture classification was used in this
study. The soils vary from silty loam to silty clay loam.
Table 3
*Soil physical properties of the Fenzidi experimental fields*

| Year | Field | Soil depth (cm) | $\theta_{fc}$ (cm$^3$/cm$^3$) | $\theta_s$ (cm$^3$/cm$^3$) | Ks (cm/d) | $\rho$ (g/cm$^3$) |
|------|-------|-----------------|-------|-------|-------|-------|
| 2016 | A | 0-10 | 0.31 | 0.47 | 11.65 | 1.47 |
| | | 10-30 | 0.31 | 0.47 | 11.65 | 1.47 |
| | | 30-50 | 0.32 | 0.51 | 48.71 | 1.36 |
| | | 50-70 | 0.39 | 0.44 | 17.48 | 1.39 |
| | | 70-100 | 0.41 | 0.44 | 40.54 | 1.45 |
| | B | 0-10 | 0.31 | 0.49 | 11.39 | 1.52 |
| | | 10-30 | 0.31 | 0.49 | 11.39 | 1.52 |
| | | 30-50 | 0.35 | 0.48 | 48.68 | 1.40 |
| | | 50-70 | 0.40 | 0.49 | 11.06 | 1.42 |
| | | 70-100 | 0.40 | 0.43 | 46.68 | 1.42 |
| 2017 | B1 | 0-10 | 0.36 | 0.42 | 5.18 | 1.52 |
| | | 10-30 | 0.36 | 0.46 | 5.18 | 1.52 |
| | | 30-50 | 0.35 | 0.47 | 11.92 | 1.38 |
| | | 50-70 | 0.42 | 0.48 | 4.41 | 1.37 |
| | | 70-100 | 0.21 | 0.47 | 6.23 | 1.69 |
| | B2 | 0-10 | 0.37 | 0.41 | 4.69 | 1.44 |
| | | 10-30 | 0.37 | 0.45 | 4.69 | 1.44 |
| | | 30-50 | 0.39 | 0.45 | 6.81 | 1.42 |
| | | 50-70 | 0.42 | 0.46 | 10.86 | 1.42 |
| | | 70-100 | 0.29 | 0.42 | 10.86 | 1.76 |

Note: $\theta_{fc}$ is the soil water content at -33 kPa, $\theta_s$ is the saturated soil water content, $K_s$ is the saturated hydraulic
conductivity, $\rho$ is the bulk density.



Table 4
*Soil texture of Fields A and B*

| Site | Depth (cm) | Soil type | Sand (%) (50-2000μm) | Silt (%) (2-50μm) | Clay (%) (0.01-2μm) |
|------|-----------|-----------|---------------------|-------------------|---------------------|
| A | 0-30 | silty clay loam | 5 | 75 | 2 |
| | 30-50 | silty loam | 22 | 7 | 8 |
| | 50-70 | silty clay loam | 3 | 8 | 17 |
| | 70-100 | silty loam | 39 | 57 | 4 |
| B | 0-30 | silty loam | 15 | 67 | 18 |
| | 30-50 | silty loam | 35 | 6 | 5 |
| | 50-70 | silty clay loam | 3 | 74 | 23 |
| | 70-100 | silty clay loam | 8 | 69 | 23 |

**2.3 The Shallow Aquifer - Vadose Zone surrogate model**
In developing the Shallow Aquifer - Vadose Zone surrogate model for modeling moisture contents in the
vadose zone, we followed the standards of good modeling practice by Jakeman et al. (2006). We made the model as
simple as possible, provide justification for our surrogate technique, test the surrogate model performance and
finally provide detail on the method to encourage discussion on the technique followed.
**2.3.1 Theoretical background**
For shallow groundwater (less than 3.3 m deep), the matric potential is a function of depth under equilibrium
conditions. Since the soil moisture characteristic curve for each soil is the relationship of moisture content and
matric potential, the moisture content is also a function of the depth of the water table under equilibrium conditions.
*Soil moisture characteristic curve*
There are several formulations describing the soil moisture characteristic curve (Bauters et al., 2000; Brooks
and Corey, 1964; Gupta and Larson, 1979; Haverkamp and Parlange, 1986; van Genuchten, 1980); the van
Genuchten and Brooks & Corey models are widely used in hydrological and soil sciences. Here, we selected the
Brooks and Corey model for its simplicity.
The Brooks-Corey model can be expressed as (Gardner et al., 1970a; Gardner et al., 1970b; Mccuen et al., 1981;
Williams et al., 1983).

$$S_e = \left(\frac{\varphi_m}{\varphi_b}\right)^{-\lambda} \qquad for \ |\varphi_m| > |\varphi_b| \qquad\qquad (1a)$$

$$S_e = 1 \qquad for \ |\varphi_m| \leq |\varphi_b| \qquad\qquad (1b)$$

in which $S_e$ is the effective saturation, $\varphi_b$ is the bubbling pressure (cm), $\varphi_m$ is matric potential (cm), and $\lambda$ is the pore
size distribution index. The effective saturation is defined as

$$S_e = \frac{\theta - \theta_d}{\theta_s - \theta_d} \qquad\qquad (2)$$

in which $\theta$ is the volumetric moisture content, $\theta_s$ is the volumetric saturated moisture content, $\theta_d$ is the residual air
dry moisture content (all in cm$^3$/cm$^3$). Equation 2 can be simplified to the form by setting $\theta_d = 0$

$$S_e = \frac{\theta}{\theta_s} \qquad\qquad (3)$$

For cases when the groundwater is close to the surface, under equilibrium conditions when the water flow is
negligible, (i.e., hydraulic potential is constant with depth), the matric potential can be expressed as height above
the water table. For our field experiment the bubbling pressure, $\varphi_b$, and the pore size distribution index, $\lambda$, in the
Brooks and Corey model can be obtained through a trial and error procedure by using the measured moisture
content and matric potential derived from the groundwater depth after an irrigation event when equilibrium state
was reached and sum of the gravity potential and matric potential was constant with depth.
**2.3.2 Parameters based on soil moisture characteristic curve**
The soil of the crop root zone is divided into several soil layers and each soil layer has its specific soil moisture
characteristic curve. After a sufficiently large irrigation and rainfall event, the moisture content is at equilibrium
after the drainage stops. After such an event, the soil moisture of vadose zone stays at the equilibrium moisture
content as long as the evapotranspiration is less than upward flux from the groundwater.
*Equilibrium moisture content*
The equilibrium soil moisture content, $\theta_{equ}$, in a layer can be determined by first replacing the matric potential
in Eq (1a) by the matric potential of the layer $\varphi_m^{z,h}$ that is dependent on the depth of the groundwater and depth of
the soil layer, z, e.g.

$$\varphi_m^{z,h} = h - z \qquad\qquad (4)$$

where $\varphi_m^{z,h}$ is the matric potential under equilibrium moisture content at a depth z below the surface and h is the
depth of the groundwater below the surface

$$\theta_{eq}^{z,h} = \theta_s^z \left(\frac{h-z}{\varphi_b^z}\right)^{-\lambda} \qquad\qquad for\ \lceil h-z\rceil > |\varphi_b^z| \qquad\qquad (5a)$$

$$\theta_{eq}^{z,h} = \theta_s^z \qquad\qquad for\ \lceil h-z\rceil \le \lceil \varphi_b^z\rceil \qquad\qquad (5b)$$

where $\theta_{eq}^{z,h}$ is the equilibrium soil moisture at the depth z below the surface while the groundwater depth is h. Note
that the superscripts *z* and *h* indicate the dependence on the distance from the soil surface, *z*, and the depth, *h*, of the
groundwater table.
***Drainable porosity***

The drainable porosity, or specific yield, is defined as the amount of water drained from the soil for a unit

decrease of the groundwater table when the soil moisture is at equilibrium. It is a crucial parameter in modeling the
moisture content in our case or amount of runoff for a shallow perched water table when there is rain (Brooks et al.,

2007).

By subtracting the total moisture content at equilibrium in the profile at the initial water table depth and at the

new position one unit lower, we obtain the drainable porosity. For example, the area between the orange and blue
curve is the amount of water drained for a decrease in the water table from 130cm to 150cm (Fig.3).

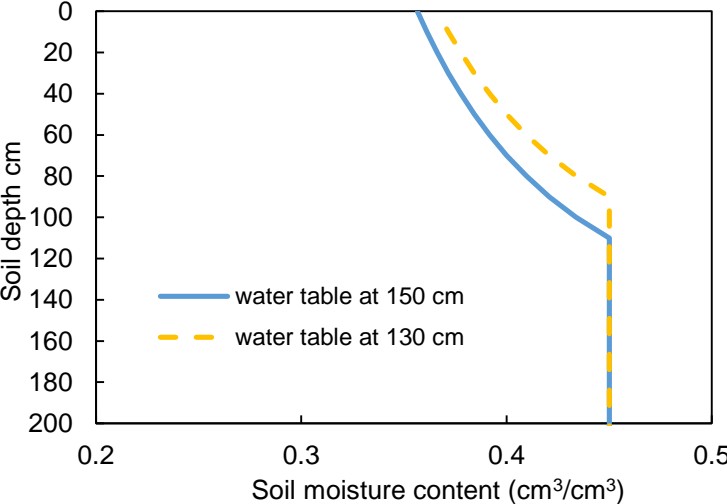


Figure. 3 Illustration of drainable porosity for a soil moisture characteristic curve with a bubbling pressure of 40 cm.
The yellow and the blue line are the equilibrium moisture contents for the groundwater depth at 130 and 150 cm,
respectively. The area between the two lines represents the amount of water for the decrease of groundwater table
drained from the profile when the groundwater decreases from 130 to 150 cm.
The total water content amount of the soil over a prescribed depth with a water table at depth $h$ can be
expressed as

$$W_{eq}^h = \sum_{j=1}^{n} L_j \overline{\left(\theta_{eq}^{z,h}\right)}_j \qquad (6)$$

where $\overline{\theta_{eq}^{z,h}}$ is the average equilibrium moisture content of layer $j$ for $h$ taken at the midpoint of the layer, $n$ is the
number of layers in the profile, $L_j$ is the height of soil layer $j$. And the drainable porosity, $\mu^h$, with the groundwater
at depth $h$, can simply be found as

$$\mu^h = \frac{W_{eq}^{h-\Delta h} - W_{eq}^{h+\Delta h}}{2\Delta h} \qquad (7)$$

where $\Delta h = 0.5 L_j$.

### 2.3.3 Calculating fluxes in the soil

The model accounts for the downward flux due to the irrigation and rainfall, evapotranspiration by plants and
soil, and upward flux from the groundwater to satisfy some or all the evapotranspiration demand by the crop and soil.
There are sets of rules implemented in an Excel spreadsheet to calculate the fluxes.
*Evapotranspiration*
1.  The plant evapotranspiration was calculated in two steps. First the daily reference evapotranspiration ($ET_0$)
was calculated by Penman-Monteith equation (Allen et al., 1998). We assumed that the moisture content
was limiting therefore the plant evapotranspiration rate was obtained by multiplying the reference
evapotranspiration by a crop coefficient (Allen et al., 1998; Sau et al., 2004; DeJonge et al., 2012). Values
for the crop coefficients were calibrated according to the water balance in the soil and found to agree with
published values for stage of crop development and soil salinity.
2.  (a) On days without rain or irrigation, the evapotranspiration lowers the water table and the moisture
content in the soil decreases due to upward movement of water to the plant roots and soil surface.
(b) On days with rain or irrigation, the potential evapotranspiration is subtracted from the irrigation and/or
rainfall and water moves downward.
*Upward flux from groundwater*
3.    The upward flux from the groundwater, $U_g^h$, is either limited by the potential evapotranspiration or the

maximum flux of groundwater. The maximum flux, $U_{g,max}^h$, depends on the depth of the groundwater, the

type of soil moisture characteristic curve, and the condition at the surface (Gardner, 1958). These equations

have an exponential form (Gardner, 1958; Yang et al., 2011; Zammouri, 2001),

$$U_{g,max}^h = \frac{a}{e^{bh} - 1} \quad \text{for } U_g^h \le ET_p \quad (8)$$

where a and b are constants and $ET_p$ is the potential evapotranspiration. The upward flux from the

groundwater can be written as:

$$U_g^h = \min\left(ET_p, U_{g,max}^h\right) \quad (9)$$

On days without rain or irrigation, the soil moisture content is calculated by taking the difference of

the equilibrium moisture content associated with the change in depth of groundwater. If in addition the

upward flux is less than evapotranspiration, the difference between the upward flux and the

evapotranspiration is extracted out of the root zone according to a predetermined distribution,$r_j$, e.g.,

$$\overline{(\theta^{z,h,t})}_j = \overline{(\theta^{z,h,t-\Delta t})}_j + \overline{(\theta_{eq}^{z,h,t})}_j - \overline{(\theta_{eq}^{z,h,t-\Delta t})}_j - \frac{r_j\left(K_c ET_p - U_g^h\right)}{L_j} \quad (10)$$

Where $\overline{(\theta^{z,h,t})}_j$ is the average soil moisture content at time $t$ of layer $j$, $\overline{(\theta_{eq}^{z,h,t})}_j$ is the average equilibrium

soil moisture content of layer $j$ when the groundwater depth is $h$ at time $t$, $K_c$ is a reduction factor of the

potential evapotranspiration for saline soil water and canopy and $r_j$ is the root function that determines the

portion of the evapotranspiration is taken up by the roots in layer $j$. The value $z$ is taken at the midpoint of

layer j. The time $t$ is expressed in days and time, $t$-$\Delta t$, is the previous day.

***The downward flux***

4.    The rules for downward flux on days with the effective rain and/or irrigation are relatively simple. If the net

flux at the surface (irrigation plus rainfall minus actual evapotranspiration) is greater than needed to bring

the soil up to equilibrium moisture content, the groundwater will be recharged and the distance to soil

surface decreases and the moisture content will be equal to the equilibrium moisture content at the new

depth.

5.    When the groundwater is not recharged, the following water balance will be calculated: the rainfall and the

irrigation are added to first layer. This layer will be brought up to the equilibrium moisture content and the

remaining water fills up the next layer to the equilibrium moisture content and so on. The calculations can

be expressed as follows:

$$\overline{(\theta^{z,h,t})}_j = min\left[\overline{\left(\theta_{eq}^{z,h,t}\right)}_j , \overline{\left(\theta^{z,h,t-\Delta t}\right)}_j + \frac{R_{j-1}\Delta t}{L_j}\right] \qquad (11)$$

where for $j \geq 2$, $R_{j-1}$ is the flux from the layer above and equals

$$R_{j+1} = R_j - \frac{\left(\overline{\left(\theta_{eq}^{z,h,t}\right)}_j - \overline{\left(\theta^{z,h,t-\Delta t}\right)}\right)L_j}{\Delta t} \qquad (12)$$

For $j=1$, $R_1$ is equal to the rainfall plus the irrigation amounts minus potential evaporation

***Groundwater table depth***

6.   The groundwater in Hetao irrigation district has a small hydraulic gradient of 0.10-0.25 ‰(Ren et al., 2016).

In addition, the soil varies from a silt loam to a clay loam (Table 4) that has saturated hydraulic

conductivity of less than 2 m/day. This means that the lateral fluxes are small compared the vertical fluxes

and can therefore neglected for the calculation of the groundwater depth. Based on this assumption, the net

change in groundwater depth, $\Delta h$, can be calculated on days without rainfall or irrigation as

$$\Delta h = \frac{U_g^h}{\mu^h} \qquad (13a)$$

and days with rain or irrigation as

$$\Delta h = -\frac{R_5}{\mu^h} \qquad (13b)$$

where the upward flux, $U_g^h$, is calculated with Eq 9, the percolation of the bottom layer $R_5$ with Eq 12 and the
drainable porosity, $\mu^h$ with Eq 7. When the groundwater is close to the surface, the drainable porosity is zero. This
would make the change in groundwater infinite. Thus, we limited the maximum decrease in groundwater after the
irrigation event to be 10-20 cm based on field observations.
**2.3.4 Model calibration and validation**
The soil moisture contents were measured from May 30[th] to September 25[th] in 2016 and 2017. Groundwater
depth was observed from June 13[th] to September 26[th] in 2016 and 2017. For the convenience of simulation, the
period of June 13[th] to September 25[th] was set as the simulation period. The model parameters were calibrated with
the 2016 data and the validation with data collected in 2017 growing seasons. Soil moisture content of the top 90 cm
(0-10 cm, 10-30 cm, 30-50 cm, 50-70 cm, 70-90 cm) and the groundwater depth were simulated for model
calibration and validation.
Relatively few parameters can be calibrated in the Shallow Aquifer-Vadose Zone Model. These are the crop
coefficients $K_c$ value, the two groundwater parameters and the root function. The other input data needed for model
were the parameters in the Brooks and Corey equation (e.g., $\theta s, \theta d, \varphi_b, \lambda$.) and were obtained by fitting the equation
to the soil moisture characteristic curve of each layer of the soil. The saturated moisture content was measured
independently as well and agreed with values obtained from the fit. Reference evapotranspiration was calculated
directly from observed meteorological data.
For better understanding the model fitting performance, statistical indicators were used to evaluate the
hydrological model goodness-of-fit (Ritter and Muñoz-Carpena, 2013). The statistical indicators including the mean
relative error (*MRE*) (Dawson et al., 2006), the root mean square error (*RMSE*) ( Abrahart and See, 2000; Bowden et
al., 2002), the Nash-Sutcliffe efficiency coefficient (*NSE*) (Nash and Suscliff, 1970), the regression coefficient (*b*)
(Xu et al., 2015), the determination coefficient (*$R^2$*) and the regression slope (Krause et al., 2005)were used to
qualify the model fitting performance during the model calibration and validation in this study. These statistical
indicators can be expressed as follows:
$$MRE = \frac{1}{N}\sum_{i=1}^{N}\frac{(P_i - O_i)}{O_i} * 100\% \qquad (14)$$

$$RMSE = \sqrt{\frac{1}{N}\sum_{i=1}^{N}(P_i - O_i)^2} \qquad (15)$$

$$NSE = 1 - \frac{\sum_{i=1}^{N}(P_i - O_i)^2}{\sum_{i=1}^{N}(O_i - \overline{O})^2} \qquad (16)$$

$$b = \frac{\sum_{i=1}^{N}O_i * P_i}{\sum_{i=1}^{N}O_i^2} \qquad (17)$$

$$R^2 = \left[\frac{\sum_{i}^{N}(O_i - \overline{O})(P_i - \overline{P})}{\left[\sum_{i}^{N}(O_i - \overline{O})\right]^{0.5}\left[\sum_{i=1}^{N}(P_i - \overline{P})\right]^{0.5}}\right]^2 \qquad (18)$$

where $N$ is the total number of observations, $O_i$ and $P_i$ are the i<sup>th</sup> observed and predicted values ($i=1, 2,..., N$), and
$\bar{O}$ and $\bar{P}$ are the mean observed values and mean predicted values, respectively. For *MRE* and *RMSE*, the values
closest to 0 indicates good model predictions. NSE=1.0 means a perfect fit, and the negative NSE values indicate
that the mean observed value is a better predictor than the simulated value (Moriasi et al., 2007). For *b* and $R^2$, the
values closest to 1 indicates good model predictions.
**3 Results**
In this section, we present first the 2016 and 2017 experimental observations of the Fenzidi experimental fields
in the Hetao irrigation district (Fig.1). This is followed by the calibration and validation of the Shallow Aquifer-
Vadose Zone Model of moisture content in each of the five layers and the groundwater table depth.
**3.1 Results of the field experiment**
The total precipitation at the experimental during growing season was 62 mm in 2016 and 67 mm in 2017. The
maximum daily rainfall was 23 mm in July 2017 (Fig. 2). The reference evapotranspiration varied between 1
mm/day to 5.5 mm/day and the total $ET_0$ was 517 mm and 442 mm in the growing seasons during 2016 and 2017,
respectively (Fig.2). Daily observation consisted of groundwater depth (blue spheres, Fig.4) and soil moisture
content at five soil depths up to 90 cm (blue spheres, Fig.5) and for Fields A and B in 2016 and Fields B1 and B2 in

2017.

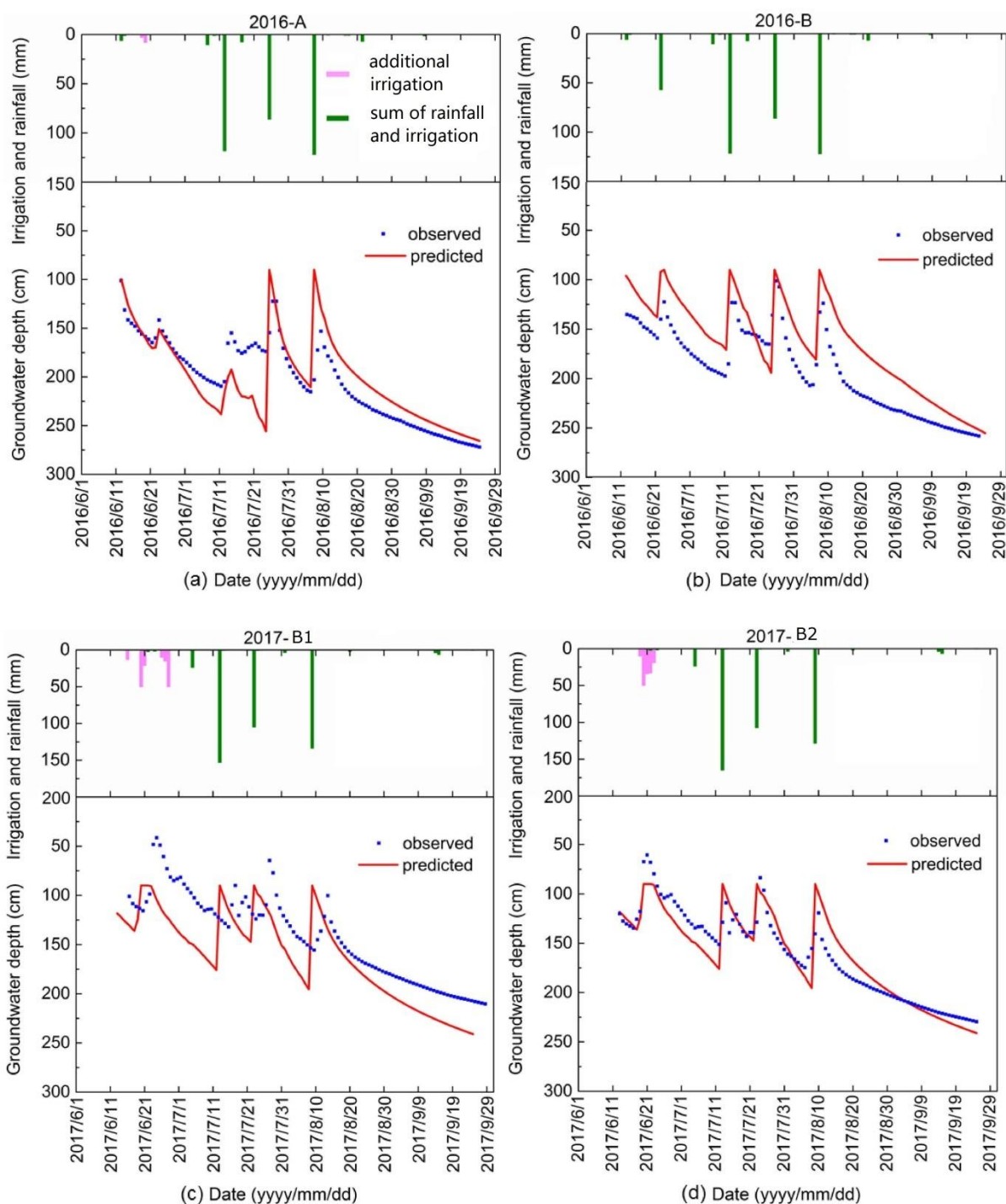


Figure.4 Simulated and observed groundwater depth during the growing period for the Fenzidi experimental fields

in the Hetao irrigation district: (a,b) calibration in 2016 and (c,d) validation in 2017. (Notes: Additional irrigation

means the irrigation recharge from the adjacent field which leads to the water table rise and was not planned).


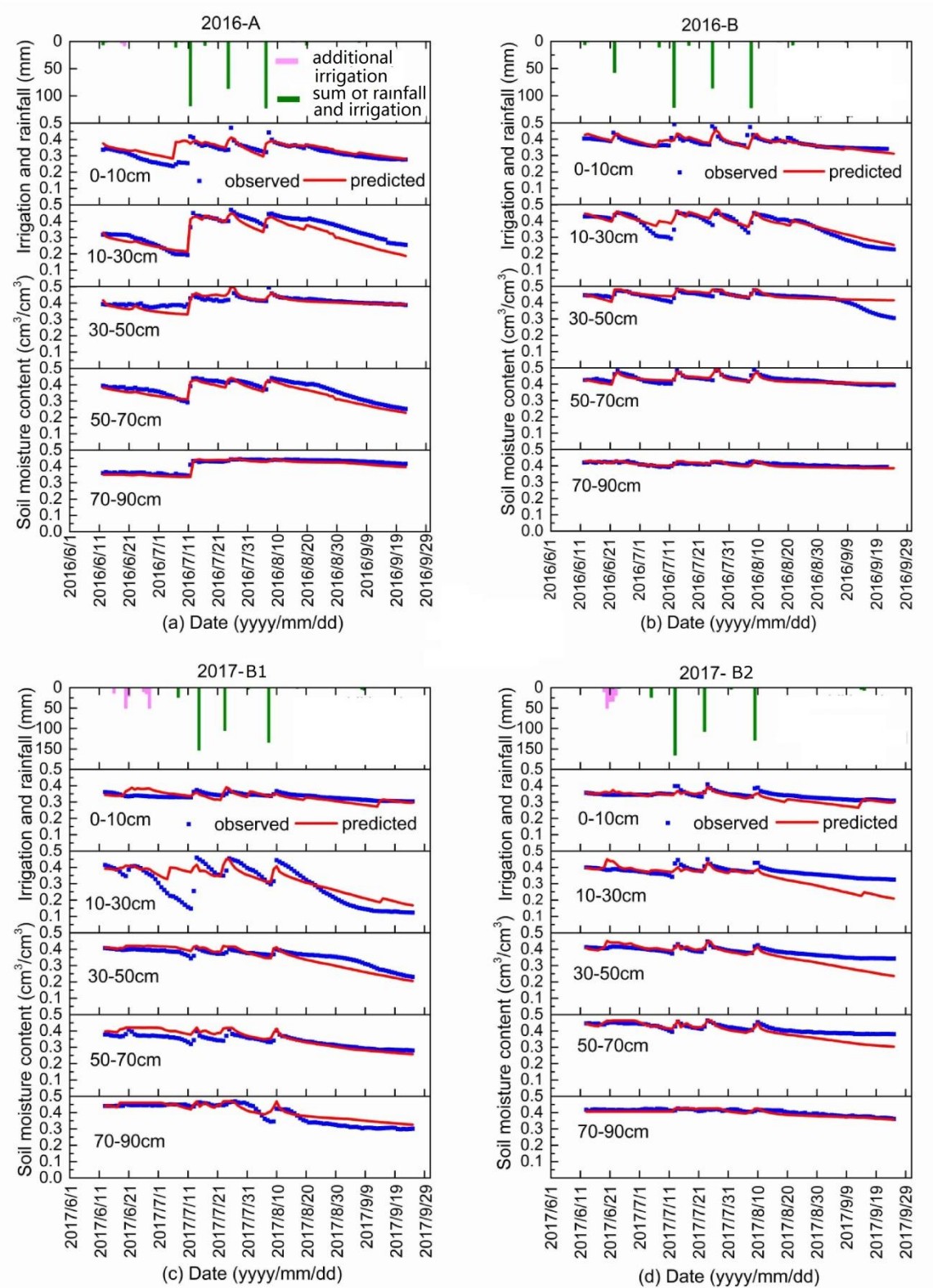



Figure. 5 Simulated and observed soil moisture content for five soil depths during the growing period for the Fenzidi
experimental fields in the Hetao irrigation district: (a, b) calibration in 2016 and (c, d) validation in 2017.
**3.1.1 Groundwater observations**
In 2016, the groundwater depth was generally more than 100 cm except during the last two irrigation events on
Field B when it reached a depth of 72 cm for one or two days (Fig. 4). In 2017, groundwater tables were slightly
closer to the surface than in 2016, especially in Field B2. The minimum groundwater depth was 61 cm on June 21,
2017 in Field B2 after an irrigation event.
In general, groundwater rose during an irrigation event and then decreased slowly due to upward movement of
water to the plant roots to meet the transpiration demand. However, in the beginning of the growing season, we can
see that the water table increased without an irrigation event.  This occurred on Field A on June 24, 2016 and Fields
B1 and B2 on June 20, 2017 (Fig. 4). This is curious and could be due to water originating from irrigation in a
nearby field.
The water table at the end of the period of observation on September 25, 2016 is approximately 2 m deep,
whereas on June 15, 2017, the depth decreased to around 125 cm. This is due to an irrigation application after the
crops were harvested to leach the salt from the surface to deeper in the profile bringing the water table up to near the
surface. Evapotranspiration during the winter is small but sufficient to bring the water table down. There was also a
rainfall event on June 5, 2017 of 13 mm (Fig. 2) before the water table was measured, increasing the water level.
**3.1.2 Soil Moisture**
Moisture contents are shown for the five layers and the two fields for 2016 and 2017 in Fig. 5. The moisture
contents were near saturation when irrigation water was added and subsequently decreased (Fig. 5). For example,
the soil moisture content changed in the 0-10 cm layer from 0.26 $cm^3/cm^3$ to 0.42 $cm^3/cm^3$ after the irrigation on
July 13, 2016 in Field A and then gradually decreased to 0.34 $cm^3/cm^3$. The moisture content decreased faster in the
10-30 cm depth than at any other depth for Fields A, B and B1 but not for Field B2. The moisture content in Field A
also showed a decrease at the 50-70 cm depth. For all plots, the moisture content at the 70-90 cm depth stayed nearly
constant and only decreased during the growing season when the water table decreased below the 150 cm depth (Fig.
5). In Field A, the initial moisture content when the observation started was less than saturation and then after the
first irrigation, remained close to the saturated moisture content.
It is interesting that while the soil profile was saturated (Fig. 4), the groundwater table was between 75-100 cm
(Fig. 5). Before equilibrium moisture content was reached the water table was likely near the surface during the
irrigation event. Because the drainable porosity was extremely small, even a minimum amount of evapotranspiration
or drainage would cause the water table to decrease to roughly the height of the capillary fringe equal to the
bubbling pressure, $\varphi_b$, in Eq. 5. The values of bubbling pressure are listed in Table 5.
**3.1.3 Soil moisture characteristic curve**
In 2016 and 2017, the observed reduced moisture contents were plotted versus the height above the water table
for the five soil layers of the two field sites in Fig. 6. These plots were used to define the soil moisture characteristic
curves which were of critical importance in simulating the moisture contents.
To define the soil moisture characteristic curve, the Brooks-Corey equation (Eq. 1) was fitted through the
points closest to saturation at each matric potential representing the equilibrium conditions after an irrigation event.
The fitted parameter values are shown in Table 5. Points to the left of the soil moisture characteristic curve are a
result of evapotranspiration drying out the soil when the upward movement of water was insufficient to replenish the
moisture content in these layers and thus matric potential and groundwater depth were not in equilibrium. In
addition, the few points to the right indicate the soil moisture was greater than the equilibrium moisture content.
Many of the outlier soil moisture contents occurred in the layer from 0-10 cm indicating that the soil was still
draining after a rainfall event shortly before the measurements. Thus, the soil was not at the equilibrium moisture
content.
The saturated moisture contents in Table 5 agree in general with the one measured in Table 1 but are not exact.
This is not a surprise as the alluvial soil deposited by the rivers with layers vary over short distances. The variation
within the field was also obvious from the soil's physical measurements. Fields B1 and B2 are within Field B. The
soil's physical properties of the various layers (Table 4) were not the same for the three sites, clearly showing the
variability within the field.
Generally, large values of pore size index coefficient λ are for sandy soils and lower values are for clay soils
(Bahmani and Bayram, 2018). We find this to be true for our site: for example, in Field A, the λ=0.23 corresponds to
a sandy layer with only 8% clay in the 30-50 cm layer (Tables 4 and 5). In the 70-90 cm layer of Field B, the λ=0.07
corresponds with the clay layer of 23% clay. In addition, bubbling pressure, $\varphi_b$, are greater for soils with a large
clay content (Bahmani and Bayram, 2018). This is demonstrated for Field A in the 10-30 cm layer where the
bubbling pressure of 75 cm corresponded with the clay layer of 20% clay. However, the correspondence between
Tables 4 and 5 is not always perfect. This is especially obvious for the layer of 70-90 cm in Field A where the values
in Table 5 clearly indicate that the soil has a dense clay layer; however, the soil description in Table 4 shows that the
soil is 39% sand. This is due to the alluvial deposition patterns with changes in soil texture over short distances as
mentioned before.

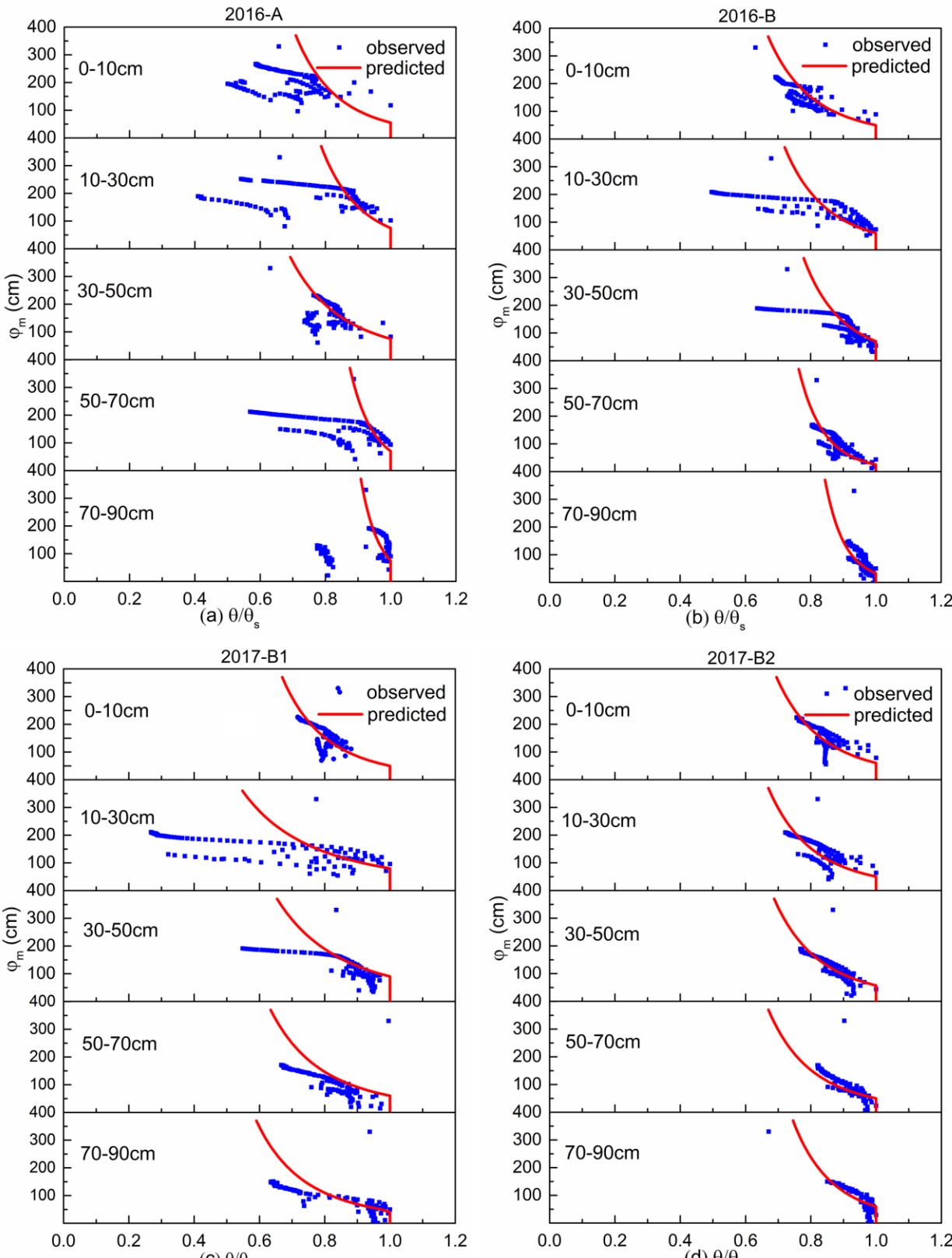


Figure.6 Soil moisture characteristic curve of the four experiment fields for the Fenzidi experimental fields. The red
line is the fit with the Brooks and Corey equation.
Table 5
*Fitted Brooks and Corey parameters for the soil moisture characteristic curve*

| Soil depth | Lamda($\lambda$) | | | | bubbling pressure ($\varphi_b$)cm | | | | saturated moisture content (cm$^3$/cm$^3$) | | | |
|---|---|---|---|---|---|---|---|---|---|---|---|---|
| Field | A | B | B1 | B2 | A | B | B1 | B2 | A | B | B1 | B2 |
| 0-10 | 0.18 | 0.2 | 0.2 | 0.2 | 55 | 50 | 50 | 60 | 0.47 | 0.49 | 0.42 | 0.41 |
| 10-30 | 0.15 | 0.18 | 0.17 | 0.2 | 75 | 60 | 70 | 50 | 0.47 | 0.48 | 0.46 | 0.45 |
| 30-50 | 0.23 | 0.15 | 0.25 | 0.2 | 75 | 70 | 50 | 57 | 0.51 | 0.48 | 0.47 | 0.45 |
| 50-70 | 0.08 | 0.1 | 0.25 | 0.2 | 70 | 25 | 30 | 50 | 0.44 | 0.49 | 0.48 | 0.46 |
| 70-90 | 0.06 | 0.07 | 0.3 | 0.16 | 75 | 33 | 45 | 59 | 0.44 | 0.43 | 0.47 | 0.42 |

3.2 Modeling results
The four parameters that can be calibrated in the Shallow Aquifer-Vadose Zone Model are the crop coefficients
$K_c$ value and the root function both related to removal of water by the atmosphere and the two groundwater
parameters that determine the upward movement of water from the groundwater.
3.2.1 Calibration of the parameters related to moisture content
The first step in the calibration was to fit the $K_c$ value from the water balance. From the moisture contents and
the groundwater depth, we can calculate approximately the amount of water lost to evapotranspiration. By
comparing these values to the reference evapotranspiration calculated with the Penman-Monteith equation, we found
that initially during the early stages the crop coefficient was 0.3 until the filling stage and then increased to 0.7
during the filling stage to the maturing stage (Table 6). These values are in accordance with the findings of Katerji et
al., (2003) that salinity reduces the evapotranspiration (Katerji et al., 2003). According to the observed total salt
content, the mean total salt content of experiment field in 0-100cm soil layer during crop growth period were
2.29g/kg in field A, 1.79g/kg in field B,  2.33g/kg in Field B1, 20.9g/kg in Field B2, respectively.
The second step was calibrating the moisture content by adapting the root function indicating from what layers
the water was taken up. Calibration was done manually by trial and error. We found that we could use the same root
function for Fields A, B, B1, and B2 (Table 6). The calibrated soil moisture contents of the five soil layers for the
two fields in general are in agreement with the measured values in 2016 (Fig 5a, b) with the coefficient of
determination $R^2$ ranging between 0.48 to 0.94 with slopes of around 1; the mean relative error (*MRE*) between -9.38%
and 6.96% and the root mean square error (*RMSE*) varied from 0.01 to 0.04 $cm^3/cm^3$ for the five layers (Table 7-1).
Finally, the parameters behaved physically realistically as water was extracted from shallow layers when the
groundwater was close to the surface and from the deeper layers when the groundwater and the associated capillary
fringe went down.
Table 6
*Calibrated parameter values of the Vadose Zone Shallow Aquifer model*

| Items | | Date | Calibrated value |
|---|---|---|---|
| Crop parameter,Kc | | June 13-July 14 | 0.3 |
| | | July 15-September 25 | 0.7 |
| Root function, $r_j$ | 0-10cm | June 13-August 7 | 0.2 |
| | | August 8-September 3 | 0.1 |
| | | September 4-October 1 | 0.1 |
| | 10-30cm | June 13-August 7 | 0.4 |
| | | August 8-September 3 | 0.4 |
| | | September 4-October 1 | 0.4 |
| | 30-50cm | June 13-August 7 | 0.3 |
| | | August 8-September 3 | 0.3 |
| | | September 4-October 1 | 0.3 |
| | 50-70cm | June 13-August 7 | 0.1 |
| | | August 8-September 3 | 0.2 |
| | | September 4-October 1 | 0.1 |
| | 70-90cm | June 13-August 7 | 0 |
| | | August 8-September 3 | 0 |
| | | September 4-October 1 | 0.1 |
| a | Field A | | 80 |
| b | | | 0.021 |
| a | Field B, B1 ,B2 | | 110 |
| b | | | 0.025 |

3.2.2. Validation of the parameters related to moisture content
The moisture contents predicted by the Shallow Aquifer-Vadose Zone Model were validated with the 2017 data
on Fields B1 and B2.  Although the validation statistics of the five layers were slightly worse than for calibration in
Table 7, the overall fit was still good as shown in Fig. 5c, d. The coefficient of determination varied between 0.39
and 0.90. The *MRE* varied between -9.34% and 19.48%, and the mean *RMSE* range was from 0.01 to 0.07 cm$^3$/cm$^3$
for the five soil layers (Table 7-2).
3.2.3 Calibration of the parameters related to groundwater depth
The final step was to calibrate the groundwater table coefficients with the 2016 data for both fields. We found
that for fields not in the same location (e.g., A, B) the subsurface was sufficiently different so that the same set of
parameters could not be used (Table 6). The difference between the calibrated parameters for the two fields was
small (Table 6). The measured and simulated groundwater depths were in good agreement with the chosen set of
parameters (Fig. 4a, b) with coefficient of determination $R^2$ being 0.67 for Field A and 0.85 for Field B (Table 7-1).
Only from July 15 to July 25 did the observed water table on Field B decrease slower than the simulated water table.
This is partly related to the fact that the properties of the soil below 90 cm were not measured, and the assumption
was made the soil moisture characteristic curve below 90 cm was the same as that from 70-90 cm. Thus the
drainable porosity of the soil which is very sensitive parameter might be different than what was used in the model.
Another reason might be that the equation for upward movement might be too simple. Other statistical indicators
show a good fit as well (Table 7-1).
Table 7-1
*Model statistics for calibration of the Shallow Aquifer model in 2016 Mean relative error, MRE; root mean square*
*error, RMSE; Regression slope; Coefficient of determination, $R^2$; Regression coefficient, b.*

| Calibration (2016) | | | | | | | | |
|---|---|---|---|---|---|---|---|---|
| | | SWC (cm$^3$/cm$^3$) | | | | | | GWD (cm) |
| | | 0-10cm | 10-30cm | 30-50cm | 50-70cm | 70-90cm | 0-90cm | |
| A | MRE(%) | 6.96 | -9.38 | -1.72 | -5.74 | -2.31 | -2.44 | -16.27 |
| | RMSE(cm$^3$/cm$^3$ or cm) | 0.04 | 0.04 | 0.02 | 0.03 | 0.01 | 0.03 | 46.52 |
| | Regression Slope | 0.51 | 0.94 | 1.34 | 1.01 | 1.05 | 0.50 | 0.50 |
| | NSE | 0.32 | 0.64 | 0.11 | 0.76 | 0.48 | 0.74 | -0.31 |
| | $R^2$ | 0.49 | 0.85 | 0.72 | 0.92 | 0.94 | 0.79 | 0.67 |
| | b | 1.05 | 0.91 | 0.99 | 0.95 | 0.98 | 0.97 | 0.81 |
| B | MRE(%) | -0.69 | 4.21 | 3.83 | -0.41 | -0.87 | 1.22 | 1.89 |
| | RMSE(cm$^3$/cm$^3$ or cm) | 0.02 | 0.03 | 0.03 | 0.01 | 0.01 | 0.02 | 18.28 |
| | Regression Slope | 0.93 | 0.72 | 0.37 | 0.76 | 1.14 | 0.76 | 0.85 |
| | NSE | 0.69 | 0.80 | 0.34 | 0.74 | -0.19 | 0.77 | 0.81 |
| | $R^2$ | 0.73 | 0.85 | 0.48 | 0.74 | 0.69 | 0.77 | 0.85 |
| | b | 0.99 | 1.03 | 1.03 | 0.99 | 0.99 | 1.00 | 1.02 |

3.2.4 Validation of the parameters related to groundwater depth

Since Fields B1 and B2 are in the same location as Field B, we used the same set of groundwater parameters

for the three fields (Table 6). The resulting fit between observed and predicted daily groundwater depths for Fields
B1 and B2 in 2017 was better than for the calibration in 2016 (Fig. 4c, d) with $R^2$ values of 0.84 for Field B1 and
0.86 for Field B2 (Table 7-2). In both cases, the slope of the regression line was close to 1. The other statistics
indicated a good fit as well (Table 7-2) with the mean relative error (*MRE*) being -0.05 for Field B1 and -0.02 for
Field B2; the root mean square error (*RMSE*) is 18.02 cm for Field B1 and 16.95 cm for Field B2; the regression
coefficient *b* is 0.94 and 1 for Fields B1 and B2, respectively. The general agreement between the measured and
simulated groundwater depth suggests that the two parameters are adequate, and the model can be used as a tool to
simulate the change of the groundwater depth.
Table 7-2
*Model statistics for validation  of the Shallow Aquifer model in 2017- Mean relative error, MRE; root mean square*
*error, RMSE; Regression slope; Coefficient of determination, $R^2$; Regression coefficient, b.*

| | | Validation (2017) | | | | | | |
|---|---|---|---|---|---|---|---|---|
| | | SWC | | | | | | GWD |
| | | 0-10cm | 10-30cm | 30-50cm | 50-70cm | 70-90cm | 0-90cm | |
| B1 | MRE(%) | -0.76 | 19.48 | -2.84 | 3.60 | 4.83 | 4.86 | -4.11 |
| | RMSE(cm$^3$/cm$^3$ or cm) | 0.02 | 0.07 | 0.03 | 0.03 | 0.03 | 0.03 | 18.02 |
| | Regression Slope | 1.03 | 0.57 | 1.38 | 1.49 | 0.70 | 0.76 | 0.80 |
| | NSE | -0.70 | 0.58 | 0.53 | 0.29 | 0.78 | 0.66 | 0.84 |
| | $R^2$ | 0.39 | 0.65 | 0.87 | 0.88 | 0.88 | 0.69 | 0.84 |
| | b | 0.99 | 1.03 | 0.99 | 1.05 | 1.03 | 1.02 | 0.94 |
| B2 | MRE(%) | -3.67 | -9.34 | -6.34 | -5.06 | -1.75 | -4.92 | 1.35 |
| | RMSE(cm$^3$/cm$^3$ or cm) | 0.02 | 0.05 | 0.04 | 0.03 | 0.01 | 0.03 | 16.95 |
| | Regression Slope | 1.11 | 1.92 | 2.24 | 1.89 | 1.02 | 1.32 | 0.94 |
| | NSE | -0.12 | -3.07 | -1.86 | -0.81 | 0.63 | 0.02 | 0.85 |
| | $R^2$ | 0.62 | 0.68 | 0.90 | 0.90 | 0.83 | 0.74 | 0.86 |
| | b | 0.96 | 0.92 | 0.95 | 0.96 | 0.98 | 0.96 | 1.00 |

**4 Discussion**

In this manuscript, a novel surrogate model was developed for irrigation systems where the groundwater is

close to the surface. The model uses the soil moisture characteristic curve to derive the drainable porosity and to
predict the moisture contents in the soil. It is based on a less often used definition of field capacity (or equilibrium
moisture content as it is called in this manuscript) based on the observation that the flow becomes negligible when
the hydraulic gradient is zero. In other words, the system is in equilibrium when the sum of the matric potential and
the gravity potential is constant. Thus, when we chose the groundwater level as the reference point for the gravity
potential, the matric potential is equal to the height above the groundwater. This is different from other application
of Darcy's law where the groundwater is below 3.3 m. In these cases, groundwater movement stops when the
conductivity becomes negligible at -33 kPa or 3.3 m in head units. The hydraulic conductivity value above -33 kPa
(3.3 m in head units) does not limit the system reaching equilibrium for daily time steps. No need therefore exists to
measure this parameter in great detail for surrogate models. The opposite is true for the soil moisture characteristic
curve for determining the spatial distribution of moisture content with depth above the groundwater.

In general, this surrogate model simulated the soil moisture content in each soil layer well, certainly when

compared to other models that attempted the soil moisture contents in the Yellow River basin such as North China
Plain (Kendy et al., 2003) and the Hetao Irrigation District by Gao et al. (2017b) during the crop growth period. Our
simulation results suggest that the reduction factor of the potential evaporation for soil saline $K_c$ and root function
parameters, together with the information of the soil moisture characteristic curves, can be used to adequately
predict the soil moisture content. To predict the groundwater depth, two additional parameters are needed for the
exponential function that defines the upward movement of groundwater.

The simulations, together with the observed data, indicated that information about the soil is very important to

obtain the exact moisture content in the soil. However, generalized soil moisture characteristic curves for each soil
type can be used in the simulation and will not result in great differences in water use by plants since percolation to
deeper layers was negligible and thus the only loss of water was by evapotranspiration independent of the soil
moisture content.

Finally, in the simulations we did not consider the influence of crop type and the influence of crop growth on

soil moisture and groundwater depth. It would be of interest to investigate in future work whether the simulations
would be improved by considering the dynamic crop characteristics during the growing season (Singh et al., 2018;
Talebizadeh et al., 2018). A mature crop model, such as the EPIC model (Williams et al., 1989) that needs
relatively few parameters, will certainly help to predict the crop yield but might not change the water use predictions.
Actually, the EPIC model already applied in Hetao irrigation district by many researchers to analyze the crop growth
during the crop growth period (Jia et al., 2012; Xu et al., 2015).

**5 Conclusion**

A novel surrogate vadose zone model for an irrigated area with a shallow aquifer was developed to simulate the fluctuation of groundwater depth and soil moisture during the crop growth stage in the shallow groundwater district. To validate and calibrate the surrogate model we carried out a two-year field experiment in the Hetao irrigation district in upper Mongolia with groundwater close to the surface. Using meteorological data and the soil moisture characteristic curve and upward capillary movement, the surrogate model predicted the soil water content with depth and groundwater height on daily time step with acceptable accuracy during validation and was an improvement two previous models applied in the Hatao district that could predict the overall water content in the root zone but not the distribution with depth.

The surrogate modeling results show that after an irrigation event as long as the upward flux from the groundwater to the root zone was greater than the plant evapotranspiration rate, the moisture contents in the vadose zone could be found directly from the soil moisture characteristic curve by equating the depth to the groundwater with the absolute value of the matric potential. When plant evapotranspiration rate exceeded the upward movement moisture contents would be indicated by groundwater depth and was predicted by a root zone function. Another finding was that the daily moisture contents were simulated without using the unsaturated hydraulic conductivity function in the surrogate model. For a daily time step equilibrium (defined as the hydraulic potential being constant) in moisture contents in the profile was attained so that precise unsaturated conductivity was not needed. Of course, for shorter time steps, predicting the transient fluxes and groundwater the conductivity function is needed. For management purposes a daily time step is acceptable.

Future improvement to this model will focus on coupling the EPIC model and apply it to simulate other crops and other location with shallow groundwater table. The surrogate model should also be compared with a "full" model, to test under what conditions the surrogate model will fall short.

**Data availability:** The observed data used in this study are not publicly accessible. These data have been collected
by personnel the College of Water Resources and Civil Engineering, China Agricultural University, with fund from
various cooperative sources. Anyone who would like to use these data, should contact Zhongyi Liu, Xingwang
Wang and Zailin Huo to obtain permission.
**Competing interests:** The authors declare that they have no conflict of interest.
**Acknowledgements:**
This study was supported by National Key Research and Development Program of China (2017YFC0403301) and
the National Natural Science Foundation of China (No. 51639009, 51679236). Peggy Stevens helped greatly with
polishing the English. We thank Xingwang Wang who helped in collecting data.

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
