# Peer review of "A Unique Vadose Zone Model for Shallow Aquifers: the Hetao Irrigation District, China Zhongyi Liu1,2, Xingwang Wang1, Zailin Huo1\*, Tammo S. Steenhuis2\* 1Center for Agricultural Water Research in China, China Agricultural University, Beijing, 100083, PR China 2Department of Biological and Environmental Engineering, Cornell University, Ithaca, NY, USA. Correspondence to: Zailin Huo (huozl@cau.edu"

_Hydrology and Earth System Sciences, 2018_

## Referee Comment (RC1) · Anonymous Referee #1 · 17 Dec 2018

The manuscript presents a new modeling approach for simulating soil moisture in the Hetao irrigation district, China, based on the soil matric potential and height of the soil layer in relation to the groundwater table. The paper fits well within the scope of a journal like HESS but there are too many issues that need to better consider by authors before it can be published. My recommendation is that the paper be sent for Major Revision, with details given below.

Major comments: 1. The introduction needs to be revised. Authors divide models based on whether they are capable of solving the full Darcy's law or whether they follow only a simplified and regionalized solution. In my opinion, such classification is not very practical making the introduction section quite confusing. On one hand, authors group very distinct models such as fully distributed catchment models, plot scale va-

[Figure]

dose zone models, and groundwater models as those based on the full solution of the Darcy's law (L82-84). On the other hand, semi-distributed catchment models are given as examples of those using simplified and regionalized solutions of the Darcy's law (L89-90). Authors should review the introduction section to focus only on similar models as theirs using comparable or alternative approaches for simulating soil moisture. 2. As a result of a confusing introduction section, it is not clear whether authors are trying to develop a model to be applied at the plot scale (which they are) or at the regional scale. Nothing is said about that in L114-118. 3. This is a clear misunderstanding of the evapotranspiration process throughout the paper, with authors referring many times simply as evaporation. Another example is given in L391 where authors refer to crop evapotranspiration (because then they refer to crop coefficients) as reference evaporation (?). 3. Soil water dynamics is pretty much dependent on soil evapotranspiration rates. However, there is nothing in the Material and Methods section describing how crop evapotranspiration is computed in the model or given as input. 4. The Material and Methods section does not detail about the approach used for calibrating/validating the model except for some vague sentence in L282-283. This information is critical and needs to be given. Not later in the results section (L385-387) when readers already gave up understanding what was done in the paper. 5. Authors apparently believe that groundwater dynamics is solely dependent on irrigation and evapotranspiration, and that groundwater flow and river connectivity are not relevant processes. This assumption seems to explain statements such as those in L328-336 which are obviously incorrect. The fact is that groundwater depth cannot be modeled using a 1D approach as in this paper, but only by considering the regional scale. Groundwater depth can only be considered as boundary condition for 1D simulations. 6. Authors assume an equilibrium between soil moisture and groundwater which does not happen in reality as themselves observed in L357-364. 7. The Conclusions section shows a brief summary of the paper, not its conclusions.

Additional comments: L49: Authors should explain why they feel water scarcity was ignored before in many parts of the world. By whom? Certainly not by population

living in those areas that have to deal daily with that problem; certainly not by the scientific community that has been addressing that problem for decades. L52: Authors give an estimate of 5100 m3 of available fresh water per capita by the year 2025. How much is it now? There is no point in advancing numbers for the future if they cannot be compared with some baseline. L56: Are these SI units? What does the "a" in "m3 a-1" stands for? Please check also other lines throughout the text (e.g. L127) L62-64: Authors should refer the environmental problems that resulted from the shallow irrigation water in Hetao, namely soil salinization risks and land degradation. L69-73: Authors should likely state that better management practices (new irrigation scheduling, alternative irrigation methods, and so on) are needed in the region. Otherwise, why the need for field trials and modeling? L74-77: One sentence does not make a paragraph. L83-84: The references for the HYDRUS and SWAP models were not given correctly. I'm sure authors of those models would appreciate seeing their work being recognized. If authors' intentions were to give applications in the Hetao region, they can be given below in the text. L92: What is the point of referring the computation method here? Are authors referring later to models using, for example, the finite volume method later? L93: The same as before. The correct reference of the HYDRUS-1D model was not given. Authors need to reword the text if their intention is to cite a modeling application. L94-96: I don't understand what authors are trying to say here. Apparently all models can be applied regardless the depth of the groundwater. L96-100: Models cited here apparently use a water bucket approach to simulate soil moisture. Is it correct? How do these fit in the model classification used in L78-79. L101-103: Why are those models not valid? Usually, water bucket approaches use empirical solutions to consider capillary rise. Couldn't those models be adapted by considering similar solutions? Apparently research in the region is quite extensive to be simply put aside. L103-107: I don't understand how the two models given here fit in the general scope of modeling research in the region. Some additional explanation should be given. L163: This should be "-33 kPa". L180: The particle size distribution is usually presented as percentage values, not fractions. L192: Equation 1 needs to

be revised. Where is $\theta$ (volumetric moisture content) and $\theta$s (volumetric saturated soil moisture content)? This text seems to be extra here. L197: The text should say "For cases... when the flow is assumed to stop..." since flow never actually stops. L201: Please revise text as it makes little sense. L237-244: Authors intention here is likely to describe the role of evapotranspiration on model computation, not evaporation. Otherwise, the assumptions are completely wrong as evaporation rates are not maximum when the plant canopy is closed. Soil evaporation is limited by the amount of energy available at the soil surface during that period in conjunction with the energy consumed by transpiration. L238-239: How is the osmotic stress considered in the model? L288: I have some doubts on whether Ren et al. (2016) is the most appropriate reference for citing statistical indicators. Did those authors develop those indicators or at least elaborated on them? Or did they simply used them like here? Please revise. L290-293: Usually, the Nash and Sutcliff modeling efficiency test is also used to assess model performance. This test allows to understand whether the residuals variance is much smaller than the observed data variance, hence that the model predictions are good. Please include it in the analysis. L30-305: This text should likely be moved to the Material and Methods section. What is the relevance of including it here to the analysis of the results? L316: Figure 4 and 5 present something defined as additional irrigation. Please explain. It does not correspond to the irrigation events given in Table 2. Also, why is it not possible to distinguish between irrigation and rainfall? Both represented by green color and during the same day. Rainfall in Figure 4 does not seem to rainfall in Figure 2. L365: I'm not sure what authors are trying to say here. Please revise. L393: Which were the salinity levels in the field? L394-395: Allen et al. (1998) does not give Kc values for soils with median salinity. Please revise. L466-467: The EPIC model was already applied to simulate crop growth in the Hetao region. Those studies should be cited.

---

## Short Comment (SC1) · 25 Dec 2018

We would like to thank Reviewer 1 for the extensive and thoughtful comments. It is obvious from the review that we have a lot of work to do in clarifying the ideas in the manuscript.  In this response, we will only touch on the major comments of the reviewer. We will provide at a later date a detailed response to all the comments.  (The numbers below are the same as the numbers in the comment of the Reviewer 1)

1. The reviewer finds our introduction confusing and does not like our division of models in those who use Darcy's law and those that do not. In the revised manuscript we will amend the divisions of the models.

   However, in this initial response, we would like to clarify the background of our division of models. In our past modelling efforts, we have noted that that making the model more complex not necessarily gives a more accurate fit to the observed data (e.g., Hoang et al., 2018, Moges et al., 2017; Steenhuis et al., 2013; Johnson et al., 2003). We show in this research that for rolling and steep terrains during periods when the precipitation exceeds the potential evaporation (and the landscape wets up), the complexities in flow patterns organize themselves in a predictable pattern of moisture contents in the landscape. The wetness pattern is a function of the amount of water stored in the watershed. The current debate in the literature is whether Darcy's law and the conservation of mass (small scale physics according to Kirchner et al., 2006) can predict these wetness pattern or we simply can use these recurring patterns to predict runoff.  We show (Steenhuis et al. 1993) for example that the runoff is linearly related with the precipitation after a threshold moisture content is exceeded. In more recent research in the Ethiopian highlands that dividing the watershed up in the periodically wet valley bottoms, degraded lands and permeable hillsides and keeping a water balance for each, we can predict the outflow more accurately  than more complex models such as SWAT and HBV (Moges et al., 2017).

   In the present paper we found something similar where the moisture content distribution after a large irrigation event depends on the ground water depth until the groundwater cannot supply the water the evaporative demand of the plants. In the present manuscript we find that we do not need the conductivity of the soil to simulate the observed moisture contents. Hence also in this case only a general form of Darcy's law is needed to model the upward movement of water.

   We agree with the reviewer that we need improve the manuscript to clarify the model division and description. Thank you for your suggestion.

2. The reviewer asks if we are developing a field or plot scale model. We are developing a field scale model that is tested in a small part of the field. We do not have the sufficient data to the do the whole field. One of the interesting results is that the soil characteristic curves determine for a large part the moisture contents in the soil as a function of the ground water depth.  Soil layering varies in the field and

so will be the moisture contents. Since our model uses the hydrologic equilibrium principle, our model remains valid at the field scale but will need information about the soil composition with depth to predict the moisture contents. Precise measurements of the moisture contents will require additional sampling. For any model to predict the spatial distribution of the moisture content with different depth will require these measurements. When average properties are taken, our and other model will predict average conditions. The next logical step in this research is to measure the soil characteristic curves with depth (and beyond the 90 cm in our current manuscript) at many locations in the fields and observe the moisture content and water table depth in the field.

3. The reviewer writes that there is "a clear misunderstanding of the evapotranspiration process throughout the paper, with authors referring many times simply as evaporation". The misunderstanding is not caused by faulty modeling of evaporation processes (some of us are modeling water balances for over 40 years!), but more likely related to the fact that we used the word "evaporation" instead of "evapotranspiration".  In the current manuscript we have followed the recommendation of Savenije (2004) who points out shortcomings in measuring transpiration due to interception and dew forming of the plants. He writes in the conclusion of his paper

> "It may be clear that I would like the word evapotranspiration to disappear from the hydrological jargon. I propose that we use the much simpler and more correct word evaporation instead. I hope that my fellow hydrologists find these arguments convincing. If not, then I look forward to a continued debate."

It looks like that we are continuing the debate.  In the rewrite we will better define what we mean with evaporation and provide in Material and Methods part of the revised manuscript a detailed account of the method that was used to calculate the evapo(transpi)ration. We will be more precise with evapo(transpi)ration terminology in the revised manuscript.

4. The reviewer points that the approach used for calibrating and validating is not detailed in the Material and Method part. We agree with the reviewer and we will give more details about the calibrating and validating process in the revised manuscript. One year was uses for calibration and one year was used for validation. To make sure that sensible representation of the moisture content was obtained we calibrated the various part of the model separately. Thanks.

5. The reviewer writes that

> "the authors apparently believe that groundwater dynamics is solely dependent on irrigation and evapotranspiration, and that groundwater flow and river connectivity are not relevant processes. This assumption seems to

explain statements such as those in L328-336 which are obviously incorrect. The fact is that groundwater depth cannot be modeled using a 1D approach as in this paper, but only by considering the regional scale".

The reviewer is correct that the groundwater is a regional phenomenon. However, the regional flows might not be the main component of the groundwater flow since the experiment takes place in a plain with a hydrologic gradient between 0.1 and 0.25% (line 124). Assuming the hydraulic conductivity is 10 m/day (It is certainly less than that since the all the soils have a high clay and silt content). This would mean a water velocity less than 5 cm/day (assuming a porosity of 0.4). The field dimensions are approximately 40 by 90 m. Consequently, it will take much longer than a year (800 days) to travel across the shortest distance, Hence, our assumption that the dynamics in the vadose zone determines the groundwater depth seems reasonable.

In spite of the argument above, we write that irrigation in a nearby field affected the groundwater table in the beginning of growing season (lines 328-336).

> "In general, groundwater rose during an irrigation event and then decreased slowly due to upward movement of water to the plant roots to meet the transpiration demand. However, in the beginning of the growing season, we can see that the water table increased without an irrigation event. This occurred on Field A on June 24, 2016 and Fields C and D on June 20, 2017 (Fig. 5). This is curious and could be due to water originating from irrigation in a nearby field."

Our hypothesis is that early in the season the cracks in the structured clays were not fully closed and these could have transported some of the water across the field. These cracks close once the field is irrigated. It is not something that can be predicted by a standard finite difference or element model since the conductivity is so small for this site. So it is unexpected (or curious).

6. The reviewer writes that

> "Authors assume an equilibrium between soil moisture and groundwater which does not happen in reality as themselves observed in L357-364."

The reviewer's comment is really helpful because the text was wrong since we did not specify that we expect equilibrium between soil moisture and groundwater after an irrigation event that causes the groundwater to rise and thus the soil is above field capacity and the hydraulic conductivity is not limiting. This equilibrium will be maintained as long as the potential upward flux is greater than the evapo(transpi)ration demand of the atmosphere. Once the calculated upward flux is less than the root function determines from what layer the "unmet" evaporation is subtracted. Our apologies for the confusion.

Reaching equilibrium (or close to it) takes one or two days according to measurement moisture content data when the soil is wet. When the soil dries out reaching equilibrium will take much longer because the unsaturated hydraulic conductivity becomes very small but not zero.

7. The reviewer points that "The Conclusions section shows a brief summary of the paper, not its conclusions". We are grateful for this useful suggestion and we will modify this part in the revised manuscript.

We will address the remaining helpful comments of the reviewer at a later date since they do not fundamentally challenge to the conceptual and theoretical part of the manuscript. We are looking forward further discussion about these excellent and major comments of reviewer 1 and other concepts in the model.

References

Hoang L., Mukundan R., Moore KEB., Owens EM and Steenhuis TS: The effect of input data resolution and complexity on the uncertainty of hydrological predictions in a humid, vegetated watershed. Hydrol. Earth Syst. Sc., 22: 5947-5965. 2018. https:// doi.org/ 10.5194/hess-22-5947-2018.

Johnson, MS., Coon, WF, Mehta, VK., Steenhuis, TS., Brooks, ES., and Boll, J.: Application of Two Hydrologic Models with Different Runoff Mechanisms to a Hillslope Dominated Watershed in the Northeastern U.S.: A Comparison of HSPF and SMR. J. Hydrol. 284:57-76. 2003. https://doi.org/10.1016/j.jhydrol.2003.07.005.

Kirchner, JW.: Getting the right answers for the right reasons: linking measurements, analyses, and models to advance the science of hydrology. Water Resour. Res., 42: W03S04. 2006. https://doi.org/10.1029/2005WR004362.

Moges, MA., Schmitter, P., Tilahun, SA., Langan, S., Dagnew, DC., Akale, AT., Steenhuis, TS.: Suitability of Watershed Models to Predict Distributed Hydrologic Response in the Awramba Watershed in the Lake Tana basin. Land Degrad. Dev, 28 (4): 1386-1397. 2017. https://doi.org/2017. 10.1002/ldr.2608.

Savenije, HHG.: The importance of interception and why we should delete the term evapotranspiration from our vocabulary. Hydrol. Process.,18: 1507–1511. https://doi.org/2004. 10.1002/hyp.5563.

Steenhuis, TS., Hrncir, M., Poteau, D., Luna, EJR., Tilahun, SA., Caballero, LA., Guzman, CD., Stoof, CR ., Sanda, M., Yitaferu, B., Cislerova, M.: A saturated excess runoff pedotransfer function for vegetated watersheds. Vadose Zone J., 12 (4). 2013. https://doi.org/10.2136/vzj2013.03.0060.

Responders: Tammo Steenhuis, Zhongyi Liu and Zailin Huo

---

## Referee Comment (RC2) · Boll (Referee) · 3 Jan 2019

This manuscript reports on field studies and model development for shallow water table dynamics following irrigation in the Yellow River basin. The results of this work will benefit water use efficiency as in this area in Inner Mongolia water use in upstream areas diminishes water available in downstream areas. The shallow-water vadose model is very straight forward, and the calibration-validation approach shows really good fits with observed data.

Major points: 1. Why does the introduction refer to Darcy type models while this manuscript does not include Darcy's law? Please clarify in the manuscript.

2. The importance of the shallow water table effects on soil moisture content is important, as this manuscript shows. Authors should refer to Brooks et al. (2007) who showed the importance of the drainable porosity to establish water table heights, and presented a similar calculation. The manuscript can emphasize more clearly the truncation of the soil moisture characteristic curve when water tables become less than 3.3m below the soil surface as part of the equilibrium moisture content calculation. (Brooks, E.S., J. Boll, and P.A. McDaniel. 2007. Distributed and integrated response of a GIS-based distributed hydrologic model. Hydrologic Processes 21:110-122.)

3. What is the reason that the fit of soil moisture is so close and the water table depths are not? Is it entirely due to soil variability or something that the model does not represent physically? Please clarify in the manuscript.

4. The manuscript includes 'additional irrigation' from an adjacent field. I assume this means water moved laterally to the study fields. This begs the question if the reverse did not also occur when the study fields were irrigated and water moved laterally to adjacent fields (some type of 'mounting' in the experimental fields). Three out of the four fields show layers with increased hydraulic conductivity, which can be responsible for such lateral movement. Please clarify.

Editorial comments: Choose 'ground water' or 'groundwater' throughout the manuscript. Line 39: change 'physical' to 'physically' (also elsewhere) Line 51-54: break up this long sentence. Line 68: change 'is' to 'will be' Line 72-73: the positive and negative effects are not clearly defined. In addition, the sentence needs rewording to: "A combination of field experiments and physically-
based modeling has the benefits of both approaches with few negative effects.
 Line 74-77: this is a single sentence paragraph without any relevant information. Line 78: suggest to change 'grouped' with 'divided' Line 79: it is not clear what is meant here with the 'full Darcy's law'. I would expect it to be the full Richards equation. – Delete 'the' Line 90: are you sure SWAT uses a regionalized Darcy's law model? Line 91: delete 'water' Line 95: why is this cutoff 3.3m? If this is related to field capacity water tension, please mention it here. Line 113: change to 'soil moisture characteristic curve' Line 125: delete 'main'

[Figure]

Line 127: check on the unit a-1 (not superscripted) as a valid metric unit for 'year' as you do later. Line 129: what is the reason to mention the number of daylight hours per year? Line 135-136: Change to 'The sowing dates were …..., respectively. Line 134: for clarity, call the fields in 2017 B1 and B2? Line 140: change 'on' to 'at' Line 142: change 'were showed' to 'are shown'; I think you mean to say 'during the growing season' because you are not identifying any growth stages explicitly in the figures. Line 143: change 'experiment' to 'experimental' Line 159: change 'crop growth period' to 'the growing season' Line 161: reword to 'soil moisture at field capacity () and at saturation (), ' Line 163: change 'measured' to 'determined' twice in this sentence. Line 166: please add texture classification to Line 168: change Table heading to 'Soil physical properties . . ..' – If fields C and D are the same as field B, what might explain the difference in soil properties shown? I suggest you add standard deviations for the average values provided. Line 180: change heading to 'Soil texture of Fields A and B' Line 188: change to 'in hydrological and soil sciences' Line 192: add comma after 'effective saturation'; note that only S and phi variables are used in this equation, so theta variables do not need to be defined. Line 196: reword (is it reasonable here to assume theta_d = 0? Figure 6 does not support this assumption. Line 201: check wording here Line 204: delete the second 'the' Line 203-206: the paragraph needs better wording; should the vadose zone stay at equilibrium moisture content instead of the groundwater? Line 209: change to 'dependent on' Figure 3: does this Figure assume a capillary fringe (bubbling pressure) of ~40cm? Maybe make note of this in the Figure caption Line 224: delete 'drained' Line 254: should the first 'and' be deleted, or is a word missing? Add 'flux' after second 'upward' Line 255: check spelling in 'prede[te]rmined' Figure 5: what explains the earlier predicted changes in groundwater depths compared to observed in 2017C and D? Line 321: the term 'additional irrigation' is not explained well here (but better in Lines 328-332). Does it mean that irrigation was applied to an adjacent field causing lateral inflow? If this is a possible effect, is there a similar lateral outflow flux possible to surrounding fields? Line 334: change 'while' to 'whereas' Line 338: switch the order of Figures 4 and 5, so they match the order of describing ground-

water and soil moisture results. Line 345: change 'at' to 'during' Line 352: Can you include the value of the bubbling pressure? Line 377: change 'indicates' to 'indicate' Line 392: add 'the' in 'to the maturing stage' Line 393: move parenthesis for the citation to just around the year (and remove the comma) Line 399: change to '. . . in general are in agreement . . .' Line 400: change 'one' to '1' Line 403: change to 'realistically' Line 408: change 'less good' to 'worse' Line 409: change to 'coefficient of determination' Line 416: change to 'depths' Line 421: no need to write out RME; change 'is' to 'being' Line 422: no need to write out RMSE Line 428: insert 'to' as in 'related to groundwater depth' Line 454: add period after 'al' Line 459: change to 'indicate' Line 466: change to 'relatively'

---

## Referee Comment (RC3) · Anonymous Referee #3 · 3 Jan 2019

Summary This manuscript describes the development of a new model that simulates soil water dynamics under shallow groundwater conditions. The model results are substantiated using field measurements. The modelling approach presented in this manuscript is good and could potentially be useful for water management purposes given its simplicity. The manuscript is well-written in general. The topic fits well within the scope of the journal. However, there are some issues that need attention before this manuscript can be considered for publication.

Major comments - L78-100: The introduction discusses about Darcy based and simplified models for soil moisture simulations. In which class does the model developed in this manuscript belong? Assuming the latter (simplified), why is this class chosen for this work? "The disadvantage is that each landscape type has a different set of regionalized landscape parameters (L88-89)" is not very clear and explicit. Please make the motivation of choosing the specific modelling approach clearer for the broad readership of the journal. - L108-113: The modelling approach in the manuscript assumes that lateral groundwater flow is negligible (i.e., groundwater dynamics is based on water input at the land surface and ET). This is a very strong assumption and should be discussed clearly in the manuscript. This is especially important because the authors mentioned "This is curious and could be due to water originating from irrigation in a nearby field (L331-332)", which gives an impression that lateral flow affects hydrology over the study area. Despite that, only vertical movement of water is considered in this study. - How is evaporation calculated? Please make that clear in Section 2. Under section 2.3.2, maximum and potential evaporation are mentioned. How are they calculated/represented? Without this information, the results presented in the manuscript are not reproducible. - The conclusion section of the manuscript is very weak. It is basically an incomplete summary of the work and fails to present the necessary elements that a conclusion section requires (e.g., usefulness and limitations). "This model is simplified, so it can be used for management purposes" is vague and does not add value.

Minor comments - I would suggest replacing physical-based with either physics-based or physically-based. - Please use "groundwater" consistently throughout the manuscript. Currently, both groundwater and ground water have been used. - L74-77: This paragraph (just one sentence!) does not fit with the previous or next one. Please re-structure and merge. - L264: "the groundwater will be recharged and increase in depth". Generally, recharge decreases the depth to groundwater table from the surface.

---

## Author Comment (AC1) · 6 Mar 2019

**Revision Notes (HESS2018581)**

**Responses to the comments of Reviewer #1:**

We would like to thank reviewer 1 for his extensive and thoughtful comments. In December 2018, we provided a general response to the comments of reviewer 1. In this document we give a detailed response to all comments repeating some of our earlier responses. Below we cite first the comment, this is followed by our response and often by a section how the text will be revised in the manuscript. The text in blue are changes and additions in the original text. For clarity we do not show any of the removed text.

**Major comments:**

**Comment** 1. The introduction needs to be revised. Authors divide models based on whether they are capable of solving the full Darcy's law or whether they follow only a simplified and regionalized solution. In my opinion, such classification is not very practical making the introduction section quite confusing. On one hand, authors group very distinct models such as fully distributed catchment models, plot scale vadose zone models, and groundwater models as those based on the full solution of the Darcy's law (L82-84). On the other hand, semi-distributed catchment models are given as examples of those using simplified and regionalized solutions of the Darcy's law (L89-90). Authors should review the introduction section to focus only on similar models as theirs using comparable or alternative approaches for simulating soil moisture.

**Response:** Thank you for your suggestion. We agree that the description of the type of models in the original models was adhoc and confusing. In the revised manuscript we follow the categorization of models proposed by Todini (2007) and Asher et al. (2015). As a consequence, we have rewritten the entire introduction. The section that relates to the model classification was changed as follows:

"There is tendency with the ever increasing computer power, to include all processes and the highly heterogeneous field conditions in hydrological models (Asher et al 2015). In case of simulating moisture contents these models become complex and often fully distributed in 3-D (Cui et al. 2017). Examples of these fully developed models are HYDRUS (Šimůnek et al., 1998), SWAP (Dam et al., 1997) and MODFLOW (Langevin et al., 2017) These models have long run times when applied to real world problems, In addition, calibration effort increases exponentially with the number of model parameters (Rosa et al., 2012; Flint et al., 2002).. This makes the use of the complex models for real time management and decision support cumbersome where many model runs are needed (Cui et al 2017).

To overcome the disadvantages of the full and completer models, computationally efficient surrogate models have been developed that speed up the modeling process without sacrificing accuracy or detail. Surrogate models are known under several names such as metamodels reduced models, model emulators, proxy models and response surfaces [e.g., Razavi et al., 2012a; Asher et al 2015]. The complex models we will call "full" or comprehensive models.

Computational efficiency is the main reason for applying surrogate models in place of full models. Other advantages of surrogate models are shortening the time needed for calibration; identifying insensitive and irrelevant parameters in the full models [Young and Ratto, 2011]; Most importantly, surrogate models allow investigating structural model uncertainty [Matott and Rabideau, 2008] Finally, surrogate models might be able to deal with better with the self- organization of complex system prevalent in hydrology than the full models (Hoang et al., 2017. For example, full models based on small scale physics (Kirchner, 2006) not necessarily can model the repetitive wetting patterns observed in humid watersheds and for that reason simple surrogate models often outperform their complex counterparts in predicting runoff when a perched water table is present in sloping terrains (Moges et al, 2017; Hoang et al 2017)

Surrogate models can be classified in two categories (Todini, 2007; Asher et al., 2015): data driven and physics derived. Data driven surrogates analyze relationships between the data available and physically derived surrogates simplify the underlying physics or reduce numerical resolution. In recent years, most emphasis in the research literature has been data driven surrogate approaches (Razavi et al. 2012a). Relatively little research has been published on physically derived approaches. Despite its popularity, data-driven surrogates can be an inefficient and unreliable approach to optimizing complex field situations especially when data is scarce such as in ground water systems (Razavi et al. 2012b) The physically derived surrogates overcome many of the limitations of data-driven approaches and are therefore superior over data driven methods (Asher et al., 2015)

**Comment** 2. As a result of a confusing introduction section, it is not clear whether authors are trying to develop a model to be applied at the plot scale (which they are) or at the regional scale. Nothing is said about that in L114-118.

**Response:** We agree that we did not address if the model was intended for the plot scale of field scale. We are developing a surrogate field scale model that is tested in a small part of the field. We do not have the sufficient data to the do the whole field. We added the following to the revised text to address this shortcoming

"The surrogate model developed is a one dimensional model simulating the moisture content in the root zone using the groundwater depth and information of soil characteristic curve. It can be easily adapted to field scale by including the lateral movement of the regional groundwater. However, in over short times, lateral movement can be neglected in nearly level areas outside a strip of 5-100 m from the river (Saleh et al., 1989) such as deltas and lakes but not over long times (Dam et al., 1997; Kendy et al, 2003)".

**Comment 3.** This is a clear misunderstanding of the evapotranspiration process throughout the paper, with authors referring many times simply as evaporation. Another example is given in L391 where authors refer to crop evapotranspiration (because then they refer to crop coefficients) as reference evaporation (?).

**Response:**

The reviewer notes that there is misunderstanding of the evapotranspiration process throughout the paper. The misunderstanding is not caused by faulty modeling of evaporation processes (some of us are modeling water balances for over 40 years!), but more likely related to the fact that we used the word "evaporation" instead of "evapotranspiration". In the current manuscript we have followed the recommendation of Savenije (2004) who points out shortcomings in measuring transpiration due to interception and dew forming of the plants. Savenije (2004) writes in the conclusion of his paper.

"It may be clear that I would like the word evapotranspiration to disappear from the hydrological jargon. I propose that we use the much simpler and more correct word evaporation instead. I hope that my fellow hydrologists find these arguments convincing. If not, then I look forward to a continued debate."

It is now obvious to us that the debate envisioned by Savenije only happened in a small group of people. Therefore, in the rewrite we have used the evapotranspiration instead of evaporation.

**Comment 4.**Soil water dynamics is pretty much dependent on soil evapotranspiration rates. However, there is nothing in the Material and Methods section describing how crop evapotranspiration is computed in the model or given as input.

**Response:** Our apologies for the oversight. We used the FAO-56 Penman-Monteith method (Allen et al., 1998) to calculate the reference crop potential evapotranspiration  $ET_0$  (mm/day). The evapotranspiration of ETp is calculated by the simplified single crop coefficient method. We calibrated the value of the crop coefficient and found as expected that it was dependent on the canopy cover and the salinity of the groundwater. We added this information in the revised manuscript as follows

"The plant evapotranspiration was calculated in two steps. First the daily reference evapotranspiration (ET0) was calculated Penman-Monteith equation (Allen et al., 1998).We assumed that the moisture content was limiting therefore the plant evaporation rate was obtained by multiplying the reference evapotranspiration by a crop coefficient. Values for the crop coefficients were calibrated according to the water balance in the soil and found to agree with published values for stage of crop development and soil salinity."

**Comment 5.** The Material and Methods section does not detail about the approach used for calibrating/validating the model except for some vague sentence in L282-283. This information is critical and needs to be given. Not later in the results section (L385-387) when readers already gave up understanding what was done in the paper.

**Response:** This is an excellent suggestion. Thanks. We moved the sentence from lines 385-387 to the material and methods section and provided in addition more details about the calibrating and validating process in the revised manuscript as follows: .

**"2.3.4 Model calibration and validation**

The soil moisture contents were measured from May 30th to September 25th in 2016 and 2017. Groundwater depth was observed from June 13th to September 26th in 2016 and 2017. For the convenience of simulation, the period of June 13th to September 25th was set as the simulation period. The model parameters were calibrated with the 2016 data

and the validation with data collected in 2017 growing seasons. Soil moisture content of the top 90 cm (0-10 cm, 10-30 cm, 30-50 cm, 50-70 cm, 70-90 cm) and the groundwater depth were simulated for model calibration and validation.

Relatively few parameters can be calibrated in the Shallow Aquifer-Vadose Zone Model. These are the crop coefficients *Kc* value, the two groundwater parameters and the root function. The other input data needed for model were the parameters in the Brooks and Corey equation (e.g.,  $\theta_s$ ,  $\theta_d$ ,  $\varphi_b$ ,  $\lambda$ .) and were obtained by fitting the equation to the soil characteristic curve of each layer of the soil. The saturated moisture content was measured indepently as well and agreed with values obtained from the fit. Reference evapotranspiration was calculated directly from observed meteorological data.

For better understanding the model fitting performance, statistical indicators were used to evaluate the hydrological model goodness-of-fit (Ritter and Muñoz-Carpena, 2013). The statistical indicators including the mean relative error (*MRE*) (Dawson et al., 2006), the root mean square error (*RMSE*, Abrahart and See, 2000; Bowden et al., 2002), the Nash-Sutcliffe efficiency coefficient (NSE, Nash and Suscliff, 1970), the regression coefficient (*b*) (Xu et al., 2015), the determination coefficient ( $R^{20}$  and the regression slope (Krause et al., 2005) were used to qualify the model fitting performance during the model calibration and validation in this study. These statistical indicators can be expressed as follows.

**Comment 6.** Authors apparently believe that groundwater dynamics is solely dependent on irrigation and evapotranspiration, and that groundwater flow and river connectivity are not relevant processes. This assumption seems to explain statements such as those in L328-336 which are obviously incorrect. The fact is that groundwater depth cannot be modeled using a 1D approach as in this paper, but only by considering the regional scale. Groundwater depth can only be considered as boundary condition for 1D simulations.

**Response:** The reviewer is correct that the groundwater is a regional phenomenon. However, the regional flows might not be the main component of the groundwater flow since the experiment takes place in a plain with a hydrologic gradient between 0.1 and 0.25% (line 124). Assuming the hydraulic conductivity is 10 m/day (It is certainly less than that since the all the soils have a high clay and silt content). This would mean a water velocity less than 5 cm/day (assuming a porosity of 0.4). The field dimensions are approximately 40 by 90 m. Consequently, it will take much more than a year (800 days) to travel across the shortest distance. We showed early in the career of the oldest author, that even in Bangladesh where the level of the rivers change over several meters between the rain and dry monsoon phase that the influence of the river was only significant in a strip of less than 100 m along the river (Saleh et al., 1989). Groundwater would rise. Hence, our assumption that the dynamics in the vadose zone determines the groundwater depth seems acceptable for the locations that are nearly level.

In spite of the argument above, we found that irrigation in a nearby field affected the groundwater table in the beginning of growing season (lines 328-336):

"In general, groundwater rose during an irrigation event and then decreased slowly due to upward movement of water to the plant roots to meet the transpiration demand. However, in the beginning of the growing season, we can see that the water table increased without an irrigation event. This occurred on Field A on June 24, 2016 and Fields C and D on June 20, 2017 (Fig. 5). This is curious and could be due to water originating from irrigation in a nearby field."

Note that Field C and D were revised as Field B1 and B2 in the revised manuscript.

One of the hypotheses of the increase in groundwater level due to irrigation in a nearby field is that early in the season the cracks in the structured clays were not fully closed and these could have transported some of the water across the field. It is not something that can be predicted by a standard finite difference or element model since the conductivity is so small for this site. So it is unexpected (or curious).

Another is that that a wetting front can proceed rapidly laterally through the root zone when the groundwater is near the surface. In this case only a very small amount of water  $\mu$  is needed to bring the soil from nearly saturated to fully saturated. It could be as little as 0.1 cm3cm-3. The wetting front velocity can then be found by v=q/ $\mu$ . Thus the wetting from can move faster by the ratio of  $\theta_s/\mu$  which could be in the order of hundreds greater than the bulk of the water. Moreover, when the soil has been plowed the conductivity of plow layer could be greater than the bulk density. So, taken both effects together, we can imagine a wetting front movement of 10-20 m/day through the root zone. Although the effect on the groundwater table is significant flux wise only a small amount of water is involved.

Since this "curious effect" only occurs with the first irrigation we believe that water movement either through cracks or root zone somehow plays an important role. Finally, we should point out that our surrogate model cannot predict it, but it is also unlikely that any "full" model will have the required equations and more importantly the input data to simulate this phenomenon.

**Comment 7.** The Conclusions section shows a brief summary of the paper, not its conclusions.

**Response:** We are grateful for this useful suggestion and we modified this part in the revised manuscript. The conclusion is formulated as:

**"5 Conclusion**

A novel surrogate vadose zone model for an irrigated area with a shallow aquifer was developed to simulate the fluctuation of groundwater depth and soil moisture during the crop growth stage in the shallow groundwater district. To validate and calibrate the surrogate model we carried out a two-year field experiment in the Hetao irrigation district in upper Mongolia with ground water close to the surface. Using meteorological data and the soil characteristic curve and upward capillary movement, the surrogate model predicted the soil water content with depth and groundwater height on daily time step with acceptable accuracy during validation and was an improvement two previous models applied in the Hatao district that could predict the overall water content in the rootzone but not the distribution with depth.

The surrogate modeling results show that after an irrigation event as long as the upward flux from the ground water to the rootzone was greater than the plant evaporation rate, the moisture contents in the vadose zone could be found directly form the soil characteristic curve by equating the depth to the groundwater with the absolute value of the matric potential. When plant evaporation rate exceeded the upward movement moisture contents became less than would be indicated by ground water depth and was predicted by a rootzone function. Another finding was that the daily moisture contents were simulated without using the unsaturated hydraulic conductivity function in the surrogate model. For a daily time step equilibrium (defined as the hydraulic potential being constant) in moisture contents in the profile was attained so that precise unsaturated conductivity was not needed. Of course, for shorter time steps, predicting the transient fluxes and groundwater the conductivity function is needed. For management purposes a daily time step is acceptable.

Future improvement to this model will focus on coupling the EPIC model and apply it to simulate other crops and other location with shallow groundwater table. The surrogate model should be also be compared with a "full" model, to test under what conditions the surrogate model will fall short."

**Additional comments:**

**Comment 1.**L49: Authors should explain why they feel water scarcity was ignored before in many parts of the world. By whom? Certainly not by population living in those areas that have to deal daily with that problem; certainly not by the scientific community that has been addressing that problem for decades.

**Response:** Here we tried to address the urgency of taking the water scarcity more seriously. It was revised as

"With global climate change and increasing human population, much of the world is facing substantial water shortage (Alcamo et al., 2007). The water crisis has caused widespread concern among public governmental officials and scientists (Guo and Shen, 2016; Oki and Kanae, 2006). Years of rapid population growth has squeezed the world water resources. The available fresh water per capita decreased 7500 m3 from 13400 m3 in 1962 to 5900 m3 in 2014 (World Bank Group, 2019)".

**Comment 2.**L52: Authors give an estimate of  $5100 \text{ m}^3$  of available fresh water per capita by the year 2025. How much is it now? There is no point in advancing numbers for the future if they cannot be compared with some baseline.

**Response:** We are grateful for your suggestion. Usually, the thresholds  $1700 \text{ m}^3$  and  $1000 \text{ m}^3$  per capita per year are used as thresholds of water stressed and water scarce, respectively. We added this information in the revised manuscript as follows:

".....Years of rapid population growth has squeezed the world water resources. The available fresh water per capita decreased 7500 m3 from 13400 m3 in 1962 to 5900 m3 in 2014 (World Bank Group, 2019).

**Comment 3.**L56: Are these SI units? What does the "a" in "m3 a-1" stands for? Please check also other lines throughout the text (e.g. L127)

**Response:** Here, "a-1" means "per annum" or "per year". "a" is the official SI unit for year (see for example: https://www.iau.org/publications/proceedings\_ruesl/units/). It is therefore being used in manuscript but we agree it is not very common. We have reverted back to "y" for year in the manuscript.

**Comment 4.**L62-64: Authors should refer the environmental problems that resulted from the shallow irrigation water in Hetao, namely soil salinization risks and land degradation.

**Response:** Thanks for your suggestion. As we know, the water from the shallow water table is a main recharge to the plant growth (Kahlown et al., 2005; Liu et al., 2016; Luo and Sophocleous, 2010). However, the salt accumulated with the upward migration of shallow groundwater table and lead to salinization (Ren et al., 2016; Yeh and Famiglietti, 2009). The Hetao district in China suffered long-term soil salinization which leads to the land degradation (Guo et al., 2018; Huang et al., 2018). This information was added in the revised manuscript. With the comment in mind we have rewritten the paragraph as:

"In the Yellow River basin, crop irrigation accounts for 96% of the total water use (Li et al., 2004). Due to the increased demand for irrigation, the river has stopped flowing downstream for an average of 70 days per year (Hinrichsen, 2002). Saving water upstream in Inner Mongolia by improved management practices means that more water will be available downstream (Gao et al., 2015). In addition, the Hetao district is suffering from salinization which leads to the land degradation (Guo et al., 2018; Huang et al., 2018). Salinization is caused by upward migration of water (and salt) from shallow groundwater table that leads to salt accumulation at the surface (Ren et al., 2016; Yeh and Famiglietti, 2009). Designing improved management practices to save water and decrease salinization can be achieved by field trials or with the aid of computer simulation mode measuring the fluxes. Field trials are time consuming, expensive and only a limited set of water management practices can be investigated. Models can test many management practices; however, the modeling results are often are questionable because they have not been validated under local field condition and have not been validated for the future conditions A combination of field experiments together with models has the benefits of both approaches."

**Comment 5.**L69-73: Authors should likely state that better management practices (new irrigation scheduling, alternative irrigation methods, and so on) are needed in the region. Otherwise, why the need for field trials and modeling?

**Response:** Please see our response to comment 4 above.

**Comment 6.**L74-77: One sentence does not make a paragraph.

**Response:** Thank you for your comment. The paragraph was amended as follows"

"Central to modeling irrigation management practices under shallow groundwater conditions (such as in the Yellow river basin) is simulating the soil moisture content accurately (Batalha et al., 2018, Gleeson et al., 2016; Jasechko and Taylor, 2015; Venkatesh et al., 2011a) because the moisture content plays a critical role in the growth of crops (Rodriguez-Iturbe, 2000), groundwater recharge (Hodnett and Bell, 1986), upward movement of water to the rootzone in areas (Gleeson et al., 2016; Jasechko and Taylor, 2015; Venkatesh et al., 2011a; Batalha et al., 2018). The latter is unique to shallow groundwater areas where the moisture content and thus the unsaturated conductivity are high and where the drying of the surface soil sets up hydraulic gradient that causes the upward capillary movement from the shallow groundwater (Kahlown et al., 2005; Liu et al., 2016; Luo and Sophocleous, 2010; Yeh and Famiglietti, 2009). The upward moving water contains salt that is deposit in the root zone and at the surface."

**Comment 7.**L83-84: The references for the HYDRUS and SWAP models were not given correctly. I'm sure authors of those models would appreciate seeing their work being recognized. If authors' intentions were to give applications in the Hetao region, they can be given below in the text.

**Response:** Apologies for the inappropriate references. References of the HYDRUS (Šimůnek et al., 1998) and SWAP (Dam et al., 1997) models were corrected in the revised manuscript. The changed text is as follows:

"There is tendency with the ever increasing computer power, to include all processes and the highly heterogeneous field conditions in hydrological models (Asher et al 2015). In case of simulating moisture contents these models become complex and often fully distributed in 3-D (Cui et al. 2017). Examples of these fully developed models are HYDRUS (Šimůnek et al., 1998), SWAP (Dam et al., 1997) and MODFLOW (Langevin et al., 2017) These models have long run times when applied to real world problems, In addition, calibration effort increases exponentially with the number of model parameters (Rosa et al., 2012; Flint et al., 2002).. This makes the use of the complex models for real time management and decision support cumbersome where many model runs are needed (Cui et al 2017). "

**Comment 8.**L92: What is the point of referring the computation method here? Are authors referring later to models using, for example, the finite volume method later?

**Response:** Thanks. We have rewritten the paragraph cited above and left out the reference to specific models. The paragraph is written as follows:

"In the Yellow River basin various models have been developed to simulate the soil water content and water fluxes. Full models that have been used are the HYDRUS-1D (Ren et al., 2016), and finite difference model application by Moiwo et al., (2010). Surrogate models for the North China plain where the groundwater is more than 20 m deep have been published by Wang et al. (2001); Kendy et al (2003); Chen et al. (2010); Ma et al. (2013); Yang et al. (2015, 2017); Li et al., (2017). In these models, the matric potential is ignored, and the hydraulic potential is equal to the gravity potential and thus

the thus the gradient of the hydraulic potential is unity (at least when it is expressed in head units). Under these conditions the water flux becomes negligible when the soil reaches field capacity at -33 KPa (equivalent to -3.3 m in head units) at what point the hydraulic conductivity becomes limiting . These models are not valid for irrigation projects along the Yellow river with shallow groundwater because the matric potential cannot be ignored over the short distance between the water table and the surface of the soil. Since the gravity and matric potential are of the same order, the water moves either down to the groundwater or up from the groundwater to the root zone depending on the matric potential at the soil (Gardner 1958; Gardener et al, 1970a,b). In summary, thus for shallow ground water at less than 3.3 m from the surface equilibrium is reached (i.e. fluxes negligible) when hydraulic gradient is zero (i.e., matric potential and gravity potential ad up to constant value) and thus not when the conductivity becomes limited at a matric potential of -33 KPa

**Comment 9.**L93: The same as before. The correct reference of the HYDRUS-1D model was not given. Authors need to reword the text if their intention is to cite a modeling application.

Response: Please see our response in comment 7 where we have cited the models correctly

**Comment 10.**L94-96: I don't understand what authors are trying to say here. Apparently all models can be applied regardless the depth of the groundwater.

**Response:** We intended to say that equilibrium is reached (i.e. fluxes stopped) when hydraulic gradient is zero (i.e., matric potential and gravity potential add up to constant value) in Darcy's law when the groundwater is close the surface at less than 3.3 m. When the groundwater is deeper than the 3.3 m the hydraulic conductivity becomes limiting before the hydraulic gradient become zero. Because it was confusing, we removed the information from the paragraph. Please see the citation of the text in the responses to comment 8 and 11.

**Comment 11.**L96-100: Models cited here apparently use a water bucket approach to simulate soil moisture. Is it correct? How do these fit in the model classification used in L78-79.

**Response:** Since all the reviewers noted that our classification was silly, we changed the classification of the models. It is now more obvious how the models are classified. The main characteristic of the surrogate model in the North China Plain with deep groundwater is that the hydraulic potential is determined by the gravity potential and thus the gradient of the hydraulic potential is unity (at least when it is expressed in head units). The models cited not necessarily assume a delta function for the hydraulic gradient (e.g. bucket model). The section reads now

"In the Yellow River basin various models have been developed to simulate the soil water content and water fluxes. Full models that have been used are the HYDRUS-1D (Ren et al., 2016), and finite difference model application by Moiwo et al., (2010). Surrogate models for the North China plain where the groundwater is more than 20 m deep have been published by Wang et al. (2001); Kendy et al (2003); Chen et al. (2010); Ma et al. (2013); Yang et al. (2015, 2017); Li et al., (2017). In these models, usually the matric potential is ignored, and the hydraulic potential is equal to the gravity potential and thus the gradient of the hydraulic potential is unity (at least when it is expressed in head units). Under these conditions the water flux becomes

negligible when the soil reaches field capacity at -33 KPa at what point the hydraulic conductivity becomes limiting. These models are not valid for irrigation projects along the Yellow river with shallow groundwater because the matric potential cannot be ignored over the short distance between the water table and the surface of the soil. Since the gravity and matric potential are of the same order, the water moves either down to the groundwater or up from the groundwater to the root zone depending on the matric potential at the soil (Gardner 1958; Grdner et al., 1970a,b). For shallow groundwater at less than 3.3 m from the surface equilibrium is reached (i.e. fluxes stopped) when hydraulic gradient is zero (i.e., matric potential and gravity potential add up to constant value) and thus not when the conductivity becomes limited at a matric potential of -33 KPa (equivalent to -3.3 m in head units)"

**Comment 12.** L101-103: Why are those models not valid? Usually, water bucket approaches use empirical solutions to consider capillary rise. Couldn't those models be adapted by considering similar solutions? Apparently research in the region is quite extensive to be simply put aside.

**Response:** Usually, for the areas with deep groundwater table, the matric potential of the soil below the root zone is ignored and thus the hydraulic potential is equal to the gravity potential. Thus the boundary condition of the root zone is free drainage. The matric potential at the groundwater is zero and therefore cannot be ignored in areas where the groundwater is close to the surface. The matric potential and the gravity potential are of the same order and depending on what the matric potential is at the surface the water moves either up or down. Please see for further detail the response to comment 11.

**Comment 13.** L103-107: I don't understand how the two models given here fit in the general scope of modeling research in the region. Some additional explanation should be given.

Response: Please see our response to comment 11 and 12. Hopefully this makes it clear.

Since this is the end of the remarks on the introduction, we have cited the rewritten introduction below. This helps to understand the various parts in the introduction relates to each other

[revised manuscript text omitted]

Comment 14. L163: This should be "-33 kPa".

**Response:** Apologies for the mistake. We corrected it as "-33kpa" in the revised manuscript.

**Comment 15.** L180: The particle size distribution is usually presented as percentage values, not fractions.

Response: We have revised it as percentage values in the revised manuscript.

| Site | Depth
(cm) | Soil type       | Sand (%)
(50-2000µm) | Silt (%)
(2-50µm) | Clay (%)
(0.01-2µm) |
|------|---------------|-----------------|-------------------------|----------------------|------------------------|
| А    | 0-30          | silty clay loam | 5                       | 75                   | 2                      |
|      | 30-50         | silty loam      | 22                      | 7                    | 8                      |
|      | 50-70         | silty clay loam | 3                       | 8                    | 17                     |
|      | 70-100        | silty loam      | 39                      | 57                   | 4                      |
| В    | 0-30          | silty loam      | 15                      | 67                   | 18                     |
|      | 30-50         | silty loam      | 35                      | 6                    | 5                      |
|      | 50-70         | silty clay loam | 3                       | 74                   | 23                     |
|      | 70-100        | silty clay loam | 8                       | 69                   | 23                     |

Table 4: Soil texture of Fields A and B

**Comment 16.**L192: Equation 1 needs to be revised. Where is  $\theta$  (volumetric moisture content) and  $\theta$ s (volumetric saturated soil moisture content)? This text seems to be extra here.

**Response:** Thanks, the text in the manuscript is revised as:

"The Brooks-Corey model can be expressed as (Gardner et al., 1970a; Gardner et al., 1970b; Mccuen et al., 1981; Williams et al., 1983).

$$S_e = \left(\frac{\varphi_m}{\varphi_b}\right)^{-\lambda} \qquad for \ |\varphi_m| > |\varphi_b| \tag{1a}$$

$$S_e = 1$$
 for  $|\varphi_m| \le |\varphi_b|$  (1b)

in which  $S_e$  is the effective saturation,  $\varphi_b$  is the bubbling pressure (cm),  $\varphi_m$  is matric potential (cm), and  $\lambda$  is the pore size distribution index. The effective saturation is defined as

$$S_e = \frac{\theta - \theta_d}{\theta_s - \theta_d} \tag{2}$$

in which  $\theta$  is the volumetric moisture content,  $\theta_s$  is the volumetric saturated moisture content,  $\theta_d$  is the residual moisture content (all in cm3/cm3). Equation 2 can be simplified to the form by setting  $\theta_d = 0$

$$S_e = \frac{\theta}{\theta_s} \tag{3}$$

For cases when the groundwater is close to the surface, under equilibrium conditions when the water flow is negligible, (i.e., hydraulic potential is constant with depth) the matric potential can be expressed as height above the water table. For our field experiment the bubbling pressure,  $\varphi_b$ , and the pore size distribution index,  $\lambda$ , in the Brooks and Corey model can be obtained through a trial and error procedure by using the measured moisture content and matric potential derived from the groundwater depth after an irrigation event when equilibrium state was reached and sum of the gravity potential and matric potential was constant with depth. "

**Comment 17.** L197: The text should say "For cases. . . when the flow is assumed to stop. . ." since flow never actually stops.

**Response:** We agree. We changed it to "when the water flow is negligible". This equivalent what was suggested to see the response to comment 16 for the change in the text

**Comment** 18.L201: Please revise text as it makes little sense.

**Response:** Hopefully our rewrite is clear. Please see the response to comment 16 for the change in the text

**Comment** 19.L237-244: Authors intention here is likely to describe the role of evapotranspiration on model computation, not evaporation. Otherwise, the assumptions are completely wrong as evaporation rates are not maximum when the plant canopy is closed. Soil evaporation is limited by the amount of energy available at the soil surface during that period in conjunction with the energy consumed by transpiration.

**Response:** That was indeed our intent. Thanks. Throughout the text, we have changed evaporation into evapotranspiration to avoid this type of confusion. The text is as follows

**Evapotranspiration**

1. The plant evapotranspiration was calculated in two steps. First the daily reference evapotranspiration (ET0) was calculated Penman-Monteith equation (Allen et al., 1998). We assumed that the moisture content was limiting therefore the plant evaporation rate was obtained by multiplying the reference evapotranspiration by a

crop coefficient. Values for the crop coefficients were calibrated according to the water balance in the soil and found to agree with published values for stage of crop development and soil salinity.

2. (a) On days without rain or irrigation, the evapotranspiration lowers the water table and the moisture content in the soil decreases due to upward movement of water to the plant roots and soil surface.

(b) On days with rain or irrigation, the potential evaporation is subtracted from the irrigation and/or rainfall and water moves downward

Comment 20. L238-239: How is the osmotic stress considered in the model?

Response: Osmotic stress is included as crop coefficient

**Comment** 21.L288: I have some doubts on whether Ren et al. (2016) is the most appropriate reference for citing statistical indicators. Did those authors develop those indicators or at least elaborated on them? Or did they simply used them like here? Please revise.

**Response:** The text is revised as follows:

"Statistical indicators were used to evaluate the hydrological model goodness-of-fit (Ritter and Muñoz-Carpena, 2013). The statistical indicators including the mean relative error (*MRE*) (Dawson et al., 2006), the root mean square error (*RMSE*, Abrahart and See, 2000; Bowden et al., 2002), the Nash-Sutcliffe efficiency coefficient (NSE, Nash and Suscliff, 1970), the regression coefficient (*b*) (Xu et al., 2015), the determination coefficient ( $R^{2}$ ) and the regression slope (Krause et al., 2005)were used to qualify the model fitting performance during the model calibration and validation in this study. These statistical indicators can be expressed as follows:

$$MRE = \frac{1}{N} \sum_{i=1}^{N} \frac{(P_i - O_i)}{O_i} * 100\% \quad (15)$$

$$NSE = 1 - \frac{\sum_{i=1}^{N} (P_i - O_i)^2}{\sum_{i=1}^{N} (O_i - \overline{O})^2}$$
(16)

$$RMSE = \sqrt{\frac{1}{N} \sum_{i=1}^{N} (P_i - O_i)^2}$$
(17)

$$B = \frac{\sum_{i=1}^{N} O_i * P_i}{\sum_{i=1}^{N} O_i^2}$$
(18)
$$R^2 = \left[ \frac{\sum_{i=1}^{N} (O_i - \overline{O})(P_i - \overline{P})}{\left[ \sum_{i=1}^{N} (O_i - \overline{O}) \right]^{0.5} \left[ \sum_{i=1}^{N} (P_i - \overline{P}) \right]^{0.5}} \right]$$
(19)

where N is the total number of observations,  $O_i$  and  $P_i$  are the ith observed and predicted values (*i*=1, 2,..., N), and  $\overline{O}$  and  $\overline{P}$  are the mean observed values and mean predicted values, respectively. For *MRE* and *RMSE*, the values closest to 0 indicate good model predictions. NSE=1.0 means a perfect fit, and the negative NSE values indicate that the mean observed value is a better predictor than the simulated value (Moriasi et al., 2007). For *b* and  $R^2$ , the values closest to 1 indicate good model prediction."

**Comment** 22.L290-293: Usually, the Nash and Sutcliff modeling efficiency test is also used to assess model performance. This test allows to understand whether the residuals variance is much smaller than the observed data variance, hence that the model predictions are good. Please include it in the analysis

**Response:** Thanks for your suggestion. The Nash and Sutcliff efficiency (NSE) is critical for the model performance and we added the value of the NSE in the revised manuscript. Please see response to comment 21 for the text in the manuscript.

**Comment** 23.L300-305: This text should likely be moved to the Material and Methods section. What is the relevance of including it here to the analysis of the results?

**Response:** In the material and method section we described how the various meteorological variables were collected. Here we describe the results of what the data indicated. The text really did not fit very well in the material and methods section and we prefer to keep it in the results section.

**Comment** 24.L316: Figure 4 and 5 present something defined as additional irrigation. Please explain. It does not correspond to the irrigation events given in Table 2. Also, why is it not possible to distinguish between irrigation and rainfall? Both represented by green color and during the same day. Rainfall in Figure 4 does not seem to rainfall in Figure 2.

**Response:** In the beginning of the growing season, the groundwater table increased without an irrigation event. This occurred on field A on June 24, 2016 and field C and D on June 20,2017 which is shown in Fig.5. This phenomenon is curious and we believe that it related to irrigation in the nearby field. Therefore, we used "additional irrigation" to simulate this increase. In the response to comment 6 we speculate on the actual causes of this phenomenon

In Figure 4 and 5, we plot the sum of the irrigation and rainfall. We changed the legend in Figure 4 and 5 to the "sum of irrigation and rainfall". Note Figure 4 was change to Figure 5 and Figure 5 was changed to Figure 4 as the Reviewer 2' suggestion for matching the order of describing groundwater and soil moisture results.

---

## Author Comment (AC2) · 6 Mar 2019

**Responses to the comments of Reviewer #2:**

We would like to thank reviewer 2 for the detailed comments. In this document we give a detailed response to all comments. Below we cite first the comment, this is followed by our response and often by a section how the text will be revised in the manuscript. The text in blue are changes and additions in the original text. For clarity we do not show any of the removed text. Thanks so much
Tammo and Zhongyi

**Major comments:**

**Comment**1. Why does the introduction refer to Darcy type models while this manuscript does not include Darcy's law? Please clarify in the manuscript.

**Response:** The intent was to make a distinction between our model and other models. However, the this and the other reviews remarked that we missed the mark. Therefore, we rewrote the introduction

In the revised manuscript, the section that relates to the model classification is as follows.

> """There is tendency with the ever increasing computer power, to include all processes and the highly heterogeneous field conditions in hydrological models (Asher et al 2015). In case of simulating moisture contents these models become complex and often fully distributed in 3-D (Cui et al. 2017). Examples of these fully developed models are HYDRUS (Šimůnek et al., 1998), SWAP (Dam et al., 1997) and MODFLOW (Langevin et al., 2017) These models have long run times when applied to real world problems, In addition, calibration effort increases exponentially with the number of model parameters (Rosa et al., 2012; Flint et al., 2002).. This makes the use of the complex models for real time management and decision support cumbersome where many model runs are needed (Cui et al 2017).

> To overcome the disadvantages of the full and completer models, computationally efficient surrogate models have been developed that speed up the modeling process without sacrificing accuracy or detail. Surrogate models are known under several names such as metamodels reduced models, model emulators, proxy models and response surfaces (e.g., Razavi et al., 2012a; Asher et al 2015). The complex models we will call "full" or comprehensive models.

> Computational efficiency is the main reason for applying surrogate models in place of full models. Other advantages of surrogate models are shortening the time needed for calibration; identifying insensitive and irrelevant parameters in the full models [Young and Ratto, 2011]; Most importantly, surrogate models allow investigating structural model uncertainty [Matott and Rabideau, 2008] Finally, surrogate models might be able to deal with better with the self- organization of complex system prevalent in hydrology than the full models (Hoang et al., 2017. For example, full models based on small scale physics (Kirchner, 2006) not necessarily can model the repetitive wetting patterns observed in humid watersheds and for that reason simple surrogate models often outperform their complex counterparts in predicting runoff when a perched water table is present in sloping terrains (Moges et al, 2017; Hoang et al 2017)

Surrogate models can be classified in two categories (Todini, 2007; Asher et al., 2015): data driven and physics derived. Data driven surrogates analyze relationships between the data available and physically derived surrogates simplify the underlying physics or reduce numerical resolution. In recent years, most emphasis in the research literature has been data driven surrogate approaches (Razavi et al. 2012a). Relatively little research has been published on physically derived approaches. Despite its popularity, data-driven surrogates can be an inefficient and unreliable approach to optimizing complex field situations especially when data is scarce such as in ground water systems (Razavi et al. 2012b) The physically derived surrogates overcome many of the limitations of data-driven approaches and are therefore superior over data driven methods (Asher et al., 2015)"

**Comment**2. The importance of the shallow water table effects on soil moisture content is important, as this manuscript shows. Authors should refer to Brooks et al. (2007) who showed the importance of the drainable porosity to establish water table heights, and presented a similar calculation. The manuscript can emphasize more clearly the truncation of the soil moisture characteristic curve when water tables become less than 3.3m below the soil surface as part of the equilibrium moisture content calculation. (Brooks, E.S., J. Boll, and P.A. McDaniel. 2007. Distributed and integrated response of a GIS-based distributed hydrologic model. Hydrologic Processes 21:110-122.)

**Response:** The Brooks et al (2007) paper is indeed very interesting. It should have been cited in our original manuscript because the approaches are very similar. There is a small difference however. We are interested in the drainable porosity due to a change in water table, while the Brooks et al. (2007) in interested in the total porosity in the soil that can be filled up before overland flow occurs.

The explanation similar to Brooks et al. (2007) but modified to the conditions with a decreasing water table is given with the description of the model.

"The drainable porosity, or specific yield, is defined as the amount of water drained from the soil for a unit decrease of the groundwater table when the soil moisture is at equilibrium. It is a crucial parameter in modeling the moisture content in our case or amount of runoff for a shallow erched water table when there is rain (Brooks et al., 2007).

By subtracting the total moisture content at equilibrium in the profile at the initial water table depth and at the new position one unit lower, we obtain the drainable porosity. For example, the area between the orange and blue curve is the amount of water drained for a decrease in the water table from 130cm to 150cm (Fig.3).

[Figure]

Figure. 3 Illustration of drainable porosity for a soil characteristic curve with a bubbling pressure of 40 cm. The yellow and the blue line are the equilibrium moisture contents for the groundwater depth at 130 and 150 cm, respectively. The area between the two lines represents the amount of water for the decrease of groundwater table drained from the profile when the groundwater decreases from 130 to 150 cm.

The total water content amount of the soil over a prescribed depth with a water table at depth $h$ can be expressed as

$$W_{eq}^h = \sum_{j=1}^n L_j \overline{\left(\theta_{eq}^{z,h}\right)}_j \qquad (6)$$

where $\overline{\theta_{eq}^{z,h}}$ is the average equilibrium moisture content of layer $j$ for $h$ taken at the midpoint of the layer, $n$ is the number of layers in the profile, $L_j$ is the height of soil layer $j$. And the drainable porosity, $\mu^h$, with the groundwater at depth $h$, can simply be found as

$$\mu^h = \frac{W_{eq}^{h-\Delta h} - W_{eq}^{h+\Delta h}}{2\Delta h} \qquad (7)$$

where $\Delta h = 0.5 L_j$."

**Comment3.** What is the reason that the fit of soil moisture is so close and the water table depths are not? Is it entirely due to soil variability or something that the model does not represent physically? Please clarify in the manuscript.

Response: One of the main problems is that the soil properties are only obtained till 90 cm. In addition the equation is likely to simple. I would be interesting if a full model can do better. The text was revised as follows:

"3.2.3 Calibration of the parameters related to groundwater depth

The final step was to calibrate the groundwater table coefficients with the 2016 data for

both fields. We found that for fields not in the same location (e.g., A, B) the subsurface was sufficiently different so that the same set of parameters could not be used (Table 6). The difference between the calibrated parameters for the two fields was small (Table 6). The measured and simulated groundwater depths were in good agreement with the chosen set of parameters (Fig. 5a, b) with coefficient of determination $R^2$ being 0.67 for Field A and 0.85 for Field B with most slopes of the regression line of around 1 (Table 7-1). Only from July 15 to July 25 did the observed water table on Field B decrease slower than the simulated water table. This is partly related to the fact that the properties of the soil below 90 cm were not measured, and the assumption was made the soil characteristic curve below 90 cm was the same as that from 70-90 cm. Thus the drainable porosity of the soil which is very sensitive parameter might be different than what was used in the model. Another reason might be that the equation for upward movement might be too simple. Other statistical indicators showed the good fit as well (Table 7-1)".

Note that Figure 4 was revised as Figure 5 and Figure 5 was revised as Figure 4 in the revised manuscript.

**Comment**4. The manuscript includes 'additional irrigation' from an adjacent field. I assume this means water moved laterally to the study fields. This begs the question if the reverse did not also occur when the study fields were irrigated and water moved laterally to adjacent fields (some type of 'mounting' in the experimental fields). Three out of the four fields show layers with increased hydraulic conductivity, which can be responsible for such lateral movement. Please clarify.

**Response:** We discovered this increase in water table without rainfall or irrigation during testing of the model. It is therefore difficult to reconstruct exactly what happened. It is indeed likely that the opposite occurred as well, however since the field was close to saturation only a small amount of water is needed to increase the water table. This might have not been noticeable on the field that was irrigated since it was only as small portion of the water applied.

As stated in the response comment 6 reviewer1: One of the hypotheses of the increase in groundwater level due to irrigation in a nearby field is that early in the season the cracks in the structured clays were not fully closed and these could have transported some of the water across the field. It is not something that can be predicted by a standard finite difference or element model since the conductivity is so small for this site. So it is unexpected (or curious).

Another is that that a wetting front can proceed rapidly laterally through the root zone when the groundwater is near the surface. In this case only a very small amount of water $\mu$ is needed to bring the soil from nearly saturated to fully saturated. It could be as little as 0.1 cm3cm-3. The wetting front velocity can then be found by v=q/$\mu$. Thus the wetting from can move faster by the ratio of $\theta$s/$\mu$ which could be in the order of hundreds greater than the bulk of the water. Moreover, when the soil has been plowed the conductivity of plow layer could be greater than the bulk density. So, taken both effects together, we can imagine a wetting front movement of 10-20 m/day through the root zone. Although the effect on the groundwater table is significant flux wise only a small amount of water is involved.

Since this "curious effect" only occurs with the first irrigation we believe that water movement either through cracks or root zone somehow plays an important role. Finally, we should point

out that our surrogate model cannot predict it, but it is also unlikely that any "full" model will have the required equations and more importantly the input data to simulate this phenomena.

**Editorial comments:**

**Comment**1. Choose 'groundwater' or 'groundwater' throughout the manuscript.

**Response:** Sorry for the inconsistent writing. It has been corrected as "groundwater" in the revised manuscript**.**

**Comment**2. Line 39: change 'physical' to 'physically' (also elsewhere)

**Response:** Thanks for your suggestion. It has been changed to "physically" in the revised manuscript**.**

**Comment**3. Line 51-54: break up this long sentence.

**Response:** Thank you for your suggestion. The long sentence was amended in the revised manuscript as

> "Years of rapid population growth has squeezed the world water resources. The available fresh water per capita decreased 7500 $m^3$ from 13400 $m^3$ in 1962 to 5900 $m^3$ in 2014 (World Bank Group, 2019).".

**Comment**4. Line 68: change 'is' to 'will be'

**Response:** We changed it to "will be" in the revised manuscript as your suggestion.

**Comment**5. Line 72-73: the positive and negative effects are not clearly defined. In addition, the sentence needs rewording to: "A combination of field experiments and physically- based modeling has the benefits of both approaches with few negative effects.

**Response:** Apologies for the unclear statement. We revised the paragraph as follows:

> "In the Yellow River basin, crop irrigation accounts for 96% of the total water use (Li et al., 2004). Due to the increased demand for irrigation, the river has stopped flowing downstream for an average of 70 days per year  (Hinrichsen, 2002). Saving water upstream in Inner Mongolia by improved management practices  means that more water will be available downstream (Gao et al., 2015).  In addition, the Hetao district is suffering from salinization which leads to the land degradation (Guo et al., 2018; Huang et al., 2018) . Salinization is caused by upward migration of water (and salt) from shallow groundwater table that leads to salt accumulation at the surface (Ren et al., 2016; Yeh and Famiglietti, 2009). Designing improved management practices  to save water and decrease salinization can be achieved by field trials or  with the aid of computer simulation mode measuring the fluxes. Field trials are time consuming, expensive and only a limited set of water management practices can be investigated. Models can test many management practices; however, the modeling results are often are questionable because they have not been validated under local field condition and have not  been validated for the future conditions A combination of field experiments together with models has the benefits of both approaches."

**Comment**6. Line 74-77: this is a single sentence paragraph without any relevant information.

**Response:** Thank you for your comment. The paragraph was amended as

"Central to modeling irrigation management practices under shallow groundwater conditions (such as in the Yellow river basin) is simulating the soil moisture content accurately (Batalha et al., 2018, Gleeson et al., 2016; Jasechko and Taylor, 2015; Venkatesh et al., 2011a) because the moisture content plays a critical role in the growth of crops (Rodriguez-Iturbe, 2000), groundwater recharge (Hodnett and Bell, 1986), upward movement of water to the rootzone in areas (Gleeson et al., 2016; Jasechko and Taylor, 2015; Venkatesh et al., 2011a; Batalha et al., 2018). The latter is unique to shallow groundwater areas where the moisture content and thus the unsaturated conductivity are high and where the drying of the surface soil sets up hydraulic gradient that causes the upward capillary movement from the shallow groundwater (Kahlown et al., 2005; Liu et al., 2016; Luo and Sophocleous, 2010; Yeh and Famiglietti, 2009). The upward moving water contains salt that is deposit in the root zone and at the surface. ."

**Comment**7. Line 78: suggest to change 'grouped' with 'divided' Line

**Response:** Thank you for your suggestion. Since we revised the introduction section, this line was deleted in the revised manuscript**.**

**Comment**8. Line 79: it is not clear what is meant here with the 'full Darcy's law'. I would expect it to be the full Richards equation. – Delete 'the'

**Response:** As stated above we rewrote the introduction. Hopefully the following is an improvement:

"There is tendency with the ever increasing computer power, to include all processes and the highly heterogeneous field conditions in hydrological models (Asher et al 2015). In case of simulating moisture contents these models become complex and often fully distributed in 3-D (Cui et al. 2017). Examples of these fully developed models are HYDRUS (Šimůnek et al., 1998), SWAP (Dam et al., 1997) and MODFLOW (Langevin et al., 2017) These models have long run times when applied to real world problems, In addition, calibration effort increases exponentially with the number of model parameters (Rosa et al., 2012; Flint et al., 2002).. This makes the use of the complex models for real time management and decision support cumbersome where many model runs are needed (Cui et al 2017).

To overcome the disadvantages of the full and completer models, computationally efficient surrogate models have been developed that speed up the modeling process without sacrificing accuracy or detail. Surrogate models are known under several names such as metamodels reduced models, model emulators, proxy models and response surfaces [e.g., Razavi et al., 2012a; Asher et al 2015]. The complex models we will call "full" or comprehensive models.

Computational efficiency is the main reason for applying surrogate models in place of full models. Other advantages of surrogate models are shortening the time needed for calibration; identifying insensitive and irrelevant parameters in the full models [Young

and Ratto, 2011];  Most importantly, surrogate models allow investigating  structural model uncertainty [Matott and Rabideau, 2008] Finally, surrogate models might be able to deal with better with the self- organization of complex system prevalent in hydrology than the full models (Hoang et al., 2017. For example, full models based on small scale physics (Kirchner, 2006) not necessarily can model the repetitive wetting patterns observed in humid watersheds and for that reason  simple surrogate models often outperform their complex counterparts in predicting runoff when a perched water table is present in sloping terrains (Moges et al, 2017; Hoang et al 2017)

**Comment**9. Line 90: are you sure SWAT uses a regionalized Darcy's law model?

**Response:** We agree that the whole section was poorly written The SWAT hydrology model is based on the water balance equation (Arnold et al., 1998). The TOPMODEL (Beven and Kirkby, 1979) and SAWT model are both mainly focused on studies in watersheds and large river basins. This study is focused on field hydrological process and we amended the narration about the model classification method in the revised manuscript. And the statement about the TOPMODEL and SWAT model was deleted in the revised manuscript.

To the question if SWAT used a regionalized Darcy Equation: In SWAT uses Darcy's law for each HRU that can be at many places in the landscape. Not sure if we can call this regionalized.

**Comment**10. Line 91: delete 'water'

**Response:** The "water" was deleted in the revised manuscript**.** Please see the response to comment 6 for the whole paragraph. Here are the specific sentences

"The latter is unique to shallow groundwater areas where the moisture content and thus the unsaturated conductivity are high and where the drying of the surface soil sets up hydraulic gradient that causes the upward capillary movement from the shallow groundwater (Kahlown et al., 2005; Liu et al., 2016; Luo and Sophocleous, 2010; Yeh and Famiglietti, 2009)."

**Comment**11. Line 95: why is this cutoff 3.3m? If this is related to field capacity water tension, please mention it here.

**Response:** Yes, it was related as indicated in the comment. The paragraph is as follows

"In the Yellow River basin various models have been developed to simulate the soil water content and water fluxes.  Full models that have been used are the  HYDRUS-1D (Ren et al., 2016), and finite difference model application by Moiwo et al., (2010). Surrogate models for the North China plain where the groundwater is more than 20 m deep have been published by Wang et al. (2001); Kendy et al (2003); Chen et al. (2010); Ma et al. (2013);  Yang et al. (2015, 2017); Li et al., (2017). In these models, the matric potential is  ignored, and the hydraulic potential is equal to the gravity potential and thus the thus the gradient of the hydraulic potential is unity (at least when it is expressed in head units). Under these conditions the water flux becomes negligible when the soil reaches field capacity at -33 KPa (equivalent to -3.3 m in head units) at what point the hydraulic conductivity becomes limiting . These models are not valid for irrigation projects along the Yellow river with shallow groundwater because the matric potential

cannot be ignored over the short distance between the water table and the surface of the soil. Since the gravity and matric potential are of the same order, the water moves either down to the groundwater or up from the groundwater to the root zone depending on the matric potential at the soil (Gardner 1958; Gardener et al, 1970a,b). In summary, thus for shallow ground water at less than 3.3 m from the surface equilibrium is reached (i.e. fluxes negligible) when hydraulic gradient is zero (i.e., matric potential and gravity potential add up to constant value) and thus not when the conductivity becomes limited at a matric potential of -33 KPa ".

**Comment**12. Line 113: change to 'soil moisture characteristic curve' .

**Response:** Thank you for your suggestion. The sentence was change to

"The moisture content at field capacity (which we call equilibrium moisture content in this manuscript) is thus a function of the groundwater depth and can be found with aid of the soil moisture characteristic curve"**.**

**Comment**13. Line 125: delete 'main'

**Response:** We deleted it in the revised manuscript**.**

**Comment14** .Line 127: check on the unit a-1 (not superscripted) as a valid metric unit for 'year' as you do later.

**Response:** "a" is the official SI unit for year (see for example https://www.iau.org/publications/proceedings_rules/units/). It is therefore being used in manuscript but we agree it is not very common. We have reverted back to "y" for year in the manuscript. The particular sentence was revised as

"The average annual precipitation is 180 mm and the annual potential evapotranspiration is 2225 mm (Luan et al., 2018)".

**Comment15.** Line 129: what is the reason to mention the number of daylight hours per year?

**Response:** We were of the opinion that it was the basic information for the study. Actually, it is not necessary, and we deleted this in the revised manuscript**.**

**Comment16.** Line 135-136: Change to 'The sowing dates were respectively.

**Response:** We revised the sentence to

"The sowing dates were April 24, 2016 and May 13, 2017".

**Comment17.** Line134: for clarity, call the fields in 2017 B1 and B2?

**Response:** Thank you for your suggestion and we changed the fields C and field D to the fields B1 and B2 in the revised manuscript**.**

**Comment18.** Line 140: change 'on' to 'at'.

**Response:** Thanks for your suggestion. We changed "on" to "at" in the revised manuscript.

**Comment19.** Line 142: change 'were showed' to 'are shown'; I think you mean to say 'during the growing season' because you are not identifying any growth stages explicitly in the figures.

**Response:** The sentence has been revised as

"Precipitation and $ET_0$ during the growing season are shown in Fig. 2" in the revised manuscript.

**Comment20.** Line143: change 'experiment' to 'experimental'

**Response:** We corrected "experiment" to "experimental" in the revised manuscript.

**Comment21.** Line 159: change 'crop growth period' to 'the growing season'

**Response:** Thanks for your suggestion. The Title of the Figure 2 was changed to

"Daily reference evapotranspiration ($ET_0$), and Precipitation during the growing season".

**Comment22.** Line 161: reword to 'soil moisture at field capacity () and at saturation ()

**Response:** We changed the "field capacity" to "soil moisture at field capacity" and "saturated soil moisture" to "soil moisture at saturation" in the revised manuscript. The text is now as follows

"Soil samples were collected in rings from the same five layers where moisture contents were measured and used for determining soil physical properties including soil moisture at field capacity ($\theta_{fc}$), soil moisture at saturation ($\theta_s$), dry bulk density ($\rho$), and saturated hydraulic conductivity ($K_s$) (Table 3). For Fields A, B, B1 and B2, the saturated hydraulic conductivity was determined by the constant head method. Field capacity was determined at -33 kPa and bulk density was determined by oven drying and dividing by the volume of the ring…"

**Comment23** Line 163: change 'measured' to 'determined' twice in this sentence.

**Response:** Thank you for your suggestion. The sentence was revised as

"For Fields A, B, B1 and B2, the saturated hydraulic conductivity was determined by the constant head method. Field capacity was determined at -33kPa and bulk density was determined by oven drying and dividing by the volume of the ring." in the revised manuscript.

**Comment24** Line 166: please add texture classification to

**Response:** The American soil texture classification was used in this study and this information was added in the revised manuscript.

**Comment25** Line 168: change Table heading to 'Soil physical properties : : :.' – If fields C and D are the same as field B, what might explain the difference in soil properties shown? I suggest you add standard deviations for the average values provided.

**Response:** The soil in the field was deposited when the Yellow River flooded and therefore variable, explain the differences in properties

The heading of the table was changed to

"Soil physical properties of the Fenzidi experimental fields" in the revised manuscript as your suggestion."

**Comment26.** Line 180: change heading to 'Soil texture of Fields A and B'

**Response:** Thank you for your suggestion and we changed the heading to

"Soil texture of Fields A and B".

**Comment27.** Line 188: change to 'in hydrological and soil sciences'

**Response:** we change in the revised manuscript, the phrase to

"in hydrological and soil sciences".

**Comment28.** Line 192: add comma after 'effective saturation'; note that only S and phi variables are used in this equation, so theta variables do not need to be defined.

**Response:** Thanks. The paragraph is as follows

"The Brooks-Corey model can be expressed as (Gardner et al., 1970a; Gardner et al., 1970b; Mccuen et al., 1981; Williams et al., 1983).

$$S_e = \left(\frac{\varphi_m}{\varphi_b}\right)^{-\lambda} \qquad for \ |\varphi_m| > |\varphi_b| \qquad (1a)$$

$$S_e = 1 \qquad for \ |\varphi_m| \leq |\varphi_b| \qquad (1b)$$

in which $S_e$ is the effective saturation $\varphi_b$ is the bubbling pressure (cm), $\varphi_m$ is matric potential (cm), and $\lambda$ is the pore size distribution index. The effective saturation is defined as

$$S_e = \frac{\theta - \theta_d}{\theta_s - \theta_d} \qquad (2)$$

in which $\theta$ is the volumetric moisture content, $\theta_s$ is the volumetric saturated moisture content, $\theta_d$ is the residual air dry moisture content (all in cm$^3$/cm$^3$). Equation 2 can be simplified to the form by setting $\theta_d = 0$

$$S_e = \frac{\theta}{\theta_s} \tag{3}$$

For cases when the groundwater is close to the surface, under equilibrium conditions when the water flow is negligible (i.e., hydraulic potential is constant with depth), the matric potential can be expressed as height above the water table. For our field experiment the bubbling pressure, $\varphi_b$, and the pore size distribution index, $\lambda$, in the Brooks and Corey model can be obtained through a trial and error procedure by using the measured moisture content and matric potential derived from the groundwater depth after an irrigation event when equilibrium state was reached and sum of the gravity potential and matric potential was constant with depth.

**Comment29.** Line 196: reword (is it reasonable here to assume theta_d = 0? Figure 6 does not support this assumption.

**Response:** $\theta_d$ is the airdry moisture content. Thus, the assumption is fine especially since we are only interested in the "wet" part of the soil characteristic curve. The words "air dry" are added the residual moisture content to clarify the meaning. See response to comment 28.

**Comment30.** Line 201: check wording here

**Response:** The changed wording is given at the end of the response to comment 28.

**Comment31.** Line 204: delete the second 'the'

**Response:** Thanks. We removed "the" as shown below

"The soil of the crop root zone is divided into several soil layers and each soil layer has its specific soil water characteristic curve. After a sufficiently large irrigation and rainfall event, the moisture content is at equilibrium after the drainage stops. After such an event, the soil moisture of vadose zone stays at the equilibrium moisture content as long as the evapotranspiration is less than upward flux from the groundwater".

**Comment32.** Line 203-206: the paragraph needs better wording; should the vadose zone stay at equilibrium moisture content instead of the groundwater?

**Response:** Hopefully we clarified the confusion in the rewrite. The changed text can be found in the response to comment 31.

**Comment33.** Line 209: change to 'dependent on' Figure 3: does this Figure assume a capillary fringe (bubbling pressure) of _40cm? Maybe make note of this in the Figure caption

**Response:** We changed "dependent of" to "dependent on" in the revised manuscript and revised the figure 3 title to

"Figure.3 Illustration of drainable porosity for a soil characteristic curve with a bubbling pressure of 40 cm. The yellow and the blue line are the equilibrium moisture contents for the groundwater depth at 130 and 150 cm, respectively. The area between the two lines

represents the amount of water for the decrease of groundwater table drained from the profile when the groundwater decreases from 130 to 150 cm"

**Comment34.** Line 224: delete 'drained'

**Response:** We deleted 'drained' in the revised manuscript.

**Comment35.** Line 254: should the first 'and' be deleted, or is a word missing? Add 'flux' after second 'upward

**Response:** Thanks for finding the mistake. The first "and" was deleted and we add "flux" in the revised manuscript**.**

**Comment36.** Line 255: check spelling in 'prede[te]rmined' Figure 5: what explains the earlier predicted changes in groundwater depths compared to observed in 2017C and D?

**Response: "**Predermined" was corrected to "predetermined" in the revised manuscript.

The honest answer is that we do not know. If the initial water table for field C (B1) would have been greater and similar to that in field D (B2) the prediction in field C would have been closer to the observed value. .

**Comment37.** Line 321: the term 'additional irrigation' is not explained well here (but better in Lines 328-332). Does it mean that irrigation was applied to an adjacent field causing lateral inflow? If this is a possible effect, is there a similar lateral outflow flux possible to surrounding fields?

**Response:** We attempted to answer this comment under comment 4. Please see that response.

**Comment38.** Line 334: change 'while' to 'whereas'

**Response:** Thank you for your suggestion and we changed "while" to "whereas" in the revised manuscript.

**Comment39.** Line338: switch the order of Figures 4 and 5, so they match the order of describing ground- water and soil moisture results.

**Response:** Thanks, we switched the order of Figures 4 and 5 in the revised manuscript.

**Comment40.** Line 345: change 'at' to 'during'

**Response:** "at" is changed to "during" as your suggestion in the revised manuscript.

**Comment41.** Line 352: Can you include the value of the bubbling pressure?

**Response**: The values of the bubbling pressure were shown in Table 5 and we added this information in this sentence in the revised manuscript.

"It is interesting that while the soil profile was saturated (Fig. 4), the groundwater table

was between 75-100 cm (Fig. 5).  Before equilibrium moisture content was reached the water table was likely near the surface during the irrigation event. Because the drainable porosity was extremely small, even a minimum amount of evaporation or drainage would cause the water table to decrease to roughly the height of the capillary fringe equal to the bubbling pressure, $\varphi_b$, in Eq. 5. The bubbling pressure are listed in Table 5.”

**Comment42.** Line 377: change 'indicates' to 'indicate'

**Response:** Done, thanks.

**Comment43.** Line 392: add 'the' in 'to the maturing stage'

**Response:** Thank you for your suggestion and we amended it to "the maturing stage" in the revised manuscript.

**Comment44.** Line 393: move parenthesis for the citation to just around the year (and remove the comma)

**Response:** We amended the phrase as your suggestion in the revised manuscript**.**

**Comment45.** Line 399: change to ': : : in general are in agreement : : :'

**Response:** Thank you for your suggestion. The sentence was revised in the revised manuscript.as

"The calibrated soil moisture contents of the five soil layers for the two fields in general are in agreement with the measured values in 2016 (Fig 4a, b)”

**Comment46.** Line 400: change 'one' to '1'

**Response:**  This is indeed an exception to the general rule**.** It is changed.

**Comment47.** Line 403: change to 'realistically'

**Response:** We changed "realistic" to "realistically" in the revised manuscript**.**

**Comment48.** Line408: change 'less good' to 'worse'

**Response:** Thanks. We made the change.

**Comment49.** Line 409: change to 'coefficient of determination'

**Response:** Thank you for your suggestion and we changed it to "the coefficient of determination" in the revised manuscript.

**Comment50.** Line 416: change to 'depths'

**Response:** We made the change. The text is now as follows:

"The moisture contents predicted by the Shallow Aquifer-Vadose Zone Model were validated with the 2017 data on Fields B1 and B2. Although the validation statistics of the five layers were slightly worse than for calibration in Table 7, the overall fit was still good as shown in Fig. 4c, d. The coefficient of determination varied between 0.39 and 0.90. The *MRE* varied between -0.09 and 0.19, and the mean *RMSE* range was from 0.01 to 0.07 cm$^3$/cm$^3$ for the five soil layers (Table 7-2)."

**Comment51.** Line 421: no need to write out RME; change 'is' to 'being'

**Response:** Thanks for your suggestion. The information about RME was deleted here. We amended the sentence as

"Others statistical indicators show a good fit as well (Table 7-1)" in the revised manuscript."

**Comment52.** Line 422: no need to write out RMSE

**Response:** We delete the sentence about the RMSE in the revised manuscript.

**Comment53.** Line 428: insert 'to' as in 'related to groundwater depth'

**Response:** We corrected the phrase as "related to the groundwater depth" in the revised manuscript**.**

**Comment54.** Line 454: add period after 'al'

**Response:** We changed the sentence as follows:

"In general, this surrogate model simulated the soil moisture content in each soil layer well, certainly when compared to other models that attempted the soil moisture contents in the Yellow River basin such as North China Plain (Kendy et al., 2003) and the Hetao Irrigation District by Gao et al. (2017b) during the crop growth period."

**Comment55.** Line 459: change to 'indicate'

**Response:** We changed "indicates" to "indicated" in the revised manuscript**.** Past tense is more appropriate.

**Comment56.** Line 466: change to 'relatively'

**Response:** We corrected it as "relatively" in the revised manuscript**.**

Thank you so much for the careful reading and all your suggestions**.**

**References:**

Arnold, J.G., Srinivasan, R., Muttiah, R.S. and Williams, J.R.: Large area hydrologic modeling and assessment part I: model development. J Am Water Resour As 34:73-89. https:// doi.org/10.1111/j.1752-1688.1998.tb05961.x.1998.

[revised manuscript text omitted]

---

## Author Comment (AC3) · 6 Mar 2019

**Responses to the comments of Reviewer #3:**

We would like to thank reviewer 3 for the detailed comments. Below we give a detailed response to all comments. We cite first the comment, this is followed by our response and often by a section how the text will be revised in the manuscript. The text in blue are changes and additions in the original text. For clarity we do not show any of the removed text.

**Major comments:**

**Comment1.** L78-100: The introduction discusses about Darcy based and simplified models for soil moisture simulations. In which class does the model developed in this manuscript belong? Assuming the latter (simplified), why is this class chosen for this work?

**Response:** The introduction seemed to have a good logic when we wrote it. At the end of the paper we conclude that the exact value of the hydraulic conductivity is irrelevant for daily predictions of moisture content in areas for shallow groundwater. In other words Darcy's law was only important for the long-term behavior of the groundwater. The idea was to convey this information about Darcy's law in the introduction, but this was obviously a bad idea given the reviewers' comments

We have, therefore, completely rewritten the introduction In the revised manuscript. The part that relates to class of the model is below

Modeling moisture contents

There is tendency with the ever increasing computer power, to include all processes and the highly heterogeneous field conditions in hydrological models (Asher et al 2015). In case of simulating moisture contents these models become complex and often fully distributed in 3-D (Cui et al. 2017). Examples of these fully developed models are HYDRUS (Šimůnek et al., 1998), SWAP (Dam et al., 1997) and MODFLOW (Langevin et al., 2017) These models have long run times when applied to real world problems, In addition, calibration effort increases exponentially with the number of model parameters (Rosa et al., 2012; Flint et al., 2002).. This makes the use of the complex models for real time management and decision support cumbersome where many model runs are needed (Cui et al 2017).

To overcome the disadvantages of the full and completer models, computationally efficient surrogate models have been developed that speed up the modeling process without sacrificing accuracy or detail. Surrogate models are known under several names such as metamodels reduced models, model emulators, proxy models and response surfaces (e.g., Razavi et al., 2012a;

Asher et al., 2015). The complex models we will call "full" or comprehensive models.

Computational efficiency is the main reason for applying surrogate models in place of full models. Other advantages of surrogate models are shortening the time needed for calibration; identifying insensitive and irrelevant parameters in the full models [Young and Ratto, 2011]; Most importantly, surrogate models allow investigating structural model uncertainty [Matott and Rabideau, 2008] Finally, surrogate models might be able to deal with better with the self-organization of complex system prevalent in hydrology than the full models (Hoang et al., 2017. For example, full models based on small scale physics (Kirchner, 2006) not necessarily can model the repetitive wetting patterns observed in humid watersheds and for that reason simple surrogate models often outperform their complex counterparts in predicting runoff when a perched water table is present in sloping terrains (Moges et al, 2017; Hoang et al 2017)

Surrogate models can be classified in two categories (Todini, 2007; Asher et al., 2015): data driven and physics derived. Data driven surrogates analyze relationships between the data available and physically derived surrogates simplify the underlying physics or reduce numerical resolution. In recent years, most emphasis in the research literature has been data driven surrogate approaches (Razavi et al. 2012a). Relatively little research has been published on physically derived approaches. Despite its popularity, data-driven surrogates can be an inefficient and unreliable approach to optimizing complex field situations especially when data is scarce such as in ground water systems (Razavi et al. 2012b) The physically derived surrogates overcome many of the limitations of data-driven approaches and are therefore superior over data driven methods (Asher et al., 2015)

In the Yellow River basin various models have been developed to simulate the soil water content and water fluxes. Full models that have been used are the HYDRUS-1D (Ren et al., 2016), and finite difference model application by Moiwo et al., (2010). Surrogate models for the North China plain where the groundwater is more than 20 m deep have been published by Wang et al. (2001); Kendy et al (2003); Chen et al. (2010); Ma et al. (2013); Yang et al. (2015, 2017); Li et al., (2017). In these models, the matric potential is ignored, and the hydraulic potential is equal to the gravity potential and thus the thus the gradient of the hydraulic potential is unity (at least when it is expressed in head units). Under these conditions the water flux becomes negligible when

the soil reaches field capacity at -33 KPa (equivalent to -3.3 m in head units) at what point the hydraulic conductivity becomes limiting . These models are not valid for irrigation projects along the Yellow river with shallow groundwater because the matric potential cannot be ignored over the short distance between the water table and the surface of the soil. Since the gravity and matric potential are of the same order, the water moves either down to the groundwater or up from the groundwater to the root zone depending on the matric potential at the soil (Gardner 1958; Gardener et al, 1970a,b). In summary, thus for shallow ground water at less than 3.3 m from the surface equilibrium is reached (i.e. fluxes negligible) when hydraulic gradient is zero (i.e., matric potential and gravity potential add up to constant value) and thus not when the conductivity becomes limited at a matric potential of -33 KPa

**Comment2.**L88-89"The disadvantage is that each landscape type has a different set of regionalized landscape parameters is not very clear and explicit. Please make the motivation of choosing the specific modelling approach clearer for the broad readership of the journal.

**Response:** We found that the soil characteristic curve and the groundwater depth determine the moisture content in the soil some times after irrigation. So, these two regional characteristics determine the value of the regionalized parameters for finding the moisture contents. Determining the two parameters that determine the upward flux from the groundwater is not simple and more research is needed how to define these parameters a priori.

**Comment3.**L108-113: The modelling approach in the manuscript assumes that lateral groundwater flow is negligible (i.e., groundwater dynamics is based on water input at the land surface and ET). This is a very strong assumption and should be discussed clearly in the manuscript. This is especially important because the authors mentioned

**Response:** It was an oversight not to include this information in the original manuscript. We added the following in section "Calculating the fluxed in the soil" in the revised manuscript.

The groundwater in Hetao irrigation district has a small hydraulic gradient of 0.10-0.25% (Ren et al., 2016). In addition, the soils vary from a silt loam to a clay loam (Table 4) that has a saturated hydraulic conductivity of less than 2 m/day. This means that the lateral fluxes are small compared the vertical fluxes and can therefore neglected for the calculation of the groundwater depth. Based on this assumption, the net change in groundwater depth, $\Delta h$, can be calculated on days without rainfall or irrigation as

$$\Delta h = \frac{U_g^h}{\mu^h} \qquad (13a)$$

and days with rain or irrigation as

$$\Delta h = -\frac{R_5}{\mu^h} \qquad (13b)$$

where the upward flux, $U_g^h$, is calculated with Eq 9, the percolation of the bottom layer $R_5$ with Eq 12 and the drainable porosity, $\mu^h$ with Eq 7…….

**Comment4.** "This is curious and could be due to water originating from irrigation in a nearby field (L331-332). Which gives an impression that lateral flow affects hydrology over the study area. Despite that, only vertical movement of water is considered in this study.

**Response:** As we explained in the last comment, the hydraulic gradient in this irrigation district is very small (0.1-0.25%). In the original manuscript, we wrote that irrigation in a nearby field affected the groundwater table in the beginning of growing season (lines 328-336).

"In general, groundwater rose during an irrigation event and then decreased slowly due to upward movement of water to the plant roots to meet the transpiration demand. However, in the beginning of the growing season, we can see that the water table increased without an irrigation event. This occurred on Field A on June 24, 2016 and Fields C and D on June 20, 2017 (Fig. 5). This is curious and could be due to water originating from irrigation in a nearby field."

One of the hypotheses of the increase in groundwater level due to irrigation in a nearby field is that early in the season the cracks in the structured clays were not fully closed and these could have transported some of the water across the field. It is not something that can be predicted by a standard finite difference or element model since the conductivity is so small for this site. So it is unexpected (or curious).

Another is that that a wetting front can proceed rapidly laterally through the root zone when the groundwater is near the surface. In this case only a very small amount of water μ is needed to bring the soil from nearly saturated to fully saturated. It could be as little as 0.1 cm3cm-3. The wetting front velocity can then be found by v=q/μ. Thus the wetting from can move faster by the ratio of θs/μ which could be in the order of hundreds greater than the bulk of the water. Moreover, when the soil has been plowed the conductivity of plow layer could be greater than the bulk density. So, taken both effects together, we can imagine a wetting front movement of 10-20 m/day through the root zone. Although the effect on the groundwater table is significant flux wise only a small amount of water is involved.

Since this "curious effect" only occurs with the first irrigation we believe that water movement either through cracks or root zone somehow plays an important role. Finally we should point out that our surrogate model cannot predict it, but it is also unlikely that any "full" model will have the required equations and more importantly the input data to simulate this phenomena.

**Comment5.**How is evaporation calculated? Please make that clear in Section 2.

**Response:** In the revised manuscript we describe how the evaporation is calculated as follows in

*Evapotranspiration*

1. The plant evapotranspiration was calculated in two steps. First the daily reference evapotranspiration ($ET_0$) was calculated Penman-Monteith equation (Allen et al., 1998). We assumed that the moisture content was limiting therefore the plant evaporation rate was obtained by multiplying the reference evapotranspiration by a crop coefficient. Values for the crop coefficients were calibrated according to the water balance in the soil and found to agree with published values for stage of crop development and soil salinity .

2. (a) On days without rain or irrigation, the evapotranspiration lowers the water table and the moisture content in the soil decreases due to upward movement of water to the plant roots and soil surface.

   (b) On days with rain or irrigation, the potential evaporation is subtracted from the irrigation and/or rainfall and water moves downward

**Comment6.**Under section 2.3.2, maximum and potential evaporation are mentioned. How are they calculated/represented? Without this information, the results presented in the manuscript are not reproducible.

**Response:** The rewrite of section 2.3.2 concerning the calculation is given in the response to the previous comment.

**Comment7.**The conclusion section of the manuscript is very weak. It is basically an incomplete summary of the work and fails to present the necessary elements that a conclusion section requires (e.g., usefulness and limitations). "This model is simplified, so it can be used for management purposes" is vague and does not add value.

**Response:** We are grateful for your suggestion. We revised the conclusion section as follows:

"A novel surrogate vadose zone model for an irrigated area with a shallow aquifer was developed to simulate the fluctuation of groundwater depth and soil moisture during the crop growth stage in the shallow groundwater district. To validate and calibrate the surrogate model we carried out a two-year field experiment in the Hetao irrigation district in upper Mongolia with groundwater close to the surface. Using meteorological data and the soil characteristic curve and upward capillary movement, the surrogate model predicted the soil water content with depth and groundwater height on daily time step with acceptable

accuracy during validation and was an improvement two previous models applied in the Hatao district that could predict the overall water content in the root zone but not the distribution with depth.

The surrogate modeling results show that after an irrigation event as long as the upward flux from the groundwater to the root zone was greater than the plant evaporation rate, the moisture contents in the vadose zone could be found directly from the soil characteristic curve by equating the depth to the groundwater with the absolute value of the matric potential. When plant evaporation rate exceeded the upward movement moisture contents became less than indicated by groundwater depth and was predicted by a root zone function.

Another finding was that the daily moisture contents were simulated without using the unsaturated hydraulic conductivity function in the surrogate model. For a daily time step equilibrium (defined as the hydraulic potential being constant) in moisture contents in the profile was attained so that precise unsaturated conductivity was not needed. Of course, for shorter time steps, predicting the transient fluxes and groundwater the conductivity function is needed. For management purposes a daily time step is acceptable"

**Minor comments**

**Comment1.** I would suggest replacing physical-based with either physics-based or physically-based.
**Response:** Thank you for your suggestion and we settled on "physically-derived" in the revised manuscript**.**

**Comment2.** Please use "groundwater" consistently throughout the manuscript. Currently, both groundwater and groundwater have been used.
**Response:** We used "groundwater" consistently in the revised manuscript.

**Comment3.** L74-77: This paragraph (just one sentence!) does not fit with the previous or next one. Please re-structure and merge.

**Response:** Thank you. The paragraph was amended as

"Central to modeling irrigation management practices under shallow groundwater conditions (such as in the Yellow river basin) is simulating the soil moisture content accurately (Batalha et al., 2018, Gleeson et al., 2016; Jasechko and Taylor, 2015; Venkatesh et al., 2011a) because the moisture content plays a critical role in the growth of crops (Rodriguez-Iturbe, 2000), groundwater recharge (Hodnett and Bell, 1986), upward movement of water to the rootzone in areas (Gleeson et al., 2016; Jasechko and Taylor, 2015; Venkatesh et al., 2011a; Batalha et al., 2018). The latter is unique to shallow groundwater areas where the moisture content and thus the unsaturated

conductivity are high and where the drying of the surface soil sets up hydraulic gradient that causes the upward capillary  movement from the shallow groundwater (Kahlown et al., 2005; Liu et al., 2016; Luo and Sophocleous, 2010; Yeh and Famiglietti, 2009). The upward moving water contains salt that is deposit in the root zone and at the surface. ”

**Comment4.**L264: “the groundwater will be recharged and increase in depth”. Generally, recharge decreases the depth to groundwater table from the surface.

**Response:** This is poorly worded.  The total depth of the groundwater is increasing. To make the writing clear, we formulated it as follows:

[revised manuscript text omitted]

---

## Author Response (AR2)

Revision Notes (HESS2018581)

June 1, 2019

Dear Professor Nunzio Romano,

Thank you for allowing us to resubmit the minor changed manuscript Hess-2018581 entitled "A Unique Vadose Zone Model for Shallow Aquifers: the Hetao Irrigation District, China". Your latest evaluation of the manuscript was as follows:

> "Your revised paper was much improved and is in a good shape now. I still suggest that some minor revision should be done accounting for the last refining comments from one of the reviewers."

Below we have replied to the comments of reviewer#1 point by point. In our response and in the revised manuscript we show in blue the changed text.

We are grateful to you and the reviewers for the comments and your time. We are looking forward to hearing from you whether additional changes are needed.

With high regard

Zalin Huo, Tammo Steenhuis and Zhongyi Liu

**Responses to the comments of Reviewer #1:**

**Comment 1.** L52 The 7500 m$^3$ can be computed from the volumes given next in the sentence. Thus, there is no point in giving redundant information. Please delete it.

**Response:** Thank you for your suggestions. In the revised manuscript, the sentence was revised as "The available fresh water per capita decreased from 13400 m$^3$ in 1962 to 5900 m$^3$ in 2014 (World Bank, 2019)" in line 52-53.

**Comment 2.** L76-80. The point being made in L76-80 is only developed in L81-90. Therefore, I suggest removing the paragraph in L80.

**Response:** On the sound advice of the reviewer, we removed L76-80 in the revised manuscript.

**Comment 3.** L96: Models can also take quite some time when applied to scenarios…

**Response:** The sentence was revised as "These models have long run times when applied to scenarios simulations for real world problems" in line 91-92 in the revised manuscript.

**Comment 4.** L108: There seems to be an extra "with" in the sentence.

**Response:** Apologies for not catching this before submission. The word "with" in the sentence was deleted in the revised manuscript. The sentence is revised as "Finally, surrogate models might be able to deal better with the self- organization of complex system prevalent in hydrology than the full models (Hoang et al., 2017)" in line 103-104 in the revised manuscript.

**Comment 5.** L123: The Yellow River Basin in general and Hetao in particular have been the focus of quite a few modelling studies. Please add more references here.

**Response:** We are grateful for your suggestion and we have added more references in the revised manuscript on lines 117-120 in the revised manuscript. It has been revised as "In the Yellow River basin various water accounting models have been developed to simulate the soil water content and water fluxes (Xu, et al., 2012; Chen et al., 2014; Xue and Ren, 2017; Yang et al., 2017; Ren et al., 2019). Numerical implementations are the finite element model HYDRUS-1D by Ren et al. (2016) and Luo and Sophocleous (2010) and a finite difference model by Moiwo et al., (2010)".

**Comment 6.** L134: Please delete "thus"

**Response:** Thanks for your suggestion. We delete "thus" in the revised manuscript. The sentence was revised as "In summary, for shallow groundwater at less than 3.3 m from the surface equilibrium is reached (i.e. fluxes negligible)….." in line 130 in the revised manuscript.

**Comment 7.** L279: References missing.

**Response:** Apologies for the references missing and we added the references in line 279 in the revised manuscript.

**References:**

Chen, L.; Feng, Q.; Li, F.; Li, C.: A bidirectional model for simulating soil water flow and salt transport under mulched drip irrigation with saline water. Agr. Water Manage., 146: 24-33. https:// doi.org/ 10.1016/j.agwat.2014.07.021

DeJonge, K., Ascough, J., Andales, A., Hansen, N., Garcia, L., Arabi, M.: Improving evapotranspiration simulations in the CERES-Maize model under limited irrigation. 115: 92-103. Agr. Water Manage., https:// doi.org/ 10.1016/j.agwat.2012.08.013. 2012.

Ren, D., Xu, X., Engel, B., Huang, Q., Xiong, Y., Huo Z., Huang, G.: Hydrological complexities in irrigated agro-ecosystems with fragmented land cover types and shallow groundwater: Insights from a distributed hydrological modeling method. Agr. Water Manage., 213:868-881. https:// doi.org/ 10.1016/j.agwat.2018.12.011. 2019.

Sau, F., Boote, K., Bostick, W., Jones, J., Minguez, M.,: Testing and improving evapotranspiration and soil water balance of the DSSAT crop models. Agron J., 96: 1243-1257. https:// doi.org/ 10.2134/agronj2004.1243. 2004.

Xu, X., Huang, G., Zhan, H., Qu, Z., Huang, Q.: Integration of SWAP and MODFLOW-2000 for modeling groundwater dynamics in shallow water table areas. J. Hydrol., 412: 170-181. https:// doi.org/ 10.1016/j.jhydrol.2011.07.002. 2012.

Xue, J. and Ren, L.: Assessing water productivity in the Hetao Irrigation District in Inner Mongolia by an agro-hydrological model. Irrigation Sci., 35:357-382. https:// doi.org/10.1007/s00271-017-0542-z. 2017.

Yang, J., Lei, H., Yang, D., Huang, M., Liu, D., Yuan X.: Impact of vegetation dynamics on hydrological processes in a semi-arid basin by using a land surface-hydrology coupled model. J. Hydrol., 551: 116-131., https:// doi.org/ 10.1016/j.jhydrol.2017.05.060. 2017